# Multi-Armed Bandits with Bounded Arm-Memory: Near-Optimal Guarantees for Best-Arm Identification and Regret Minimization

**Arnab Maiti** *
Indian Institute of Technology,
Kharagpur, India
arnabmaiti@iitkgp.ac.in

**Vishakha Patil**
Indian Institute of Science,
Bangalore, India
patilv@iisc.ac.in

**Arindam Khan**
Indian Institute of Science,
Bangalore, India
arindamkhan@iisc.ac.in

## Abstract

We study the Stochastic Multi-armed Bandit problem under bounded arm-memory. In this setting, the arms arrive in a stream, and the number of arms that can be stored in the memory at any time, is bounded. The decision-maker can only pull arms that are present in the memory. We address the problem from the perspective of two standard objectives: 1) regret minimization, and 2) best-arm identification. For *regret minimization*, we settle an important open question by showing an almost tight guarantee. We show $\Omega(T^{2/3})$ cumulative regret in expectation for single-pass algorithms for arm-memory size of $(n-1)$, where $n$ is the number of arms. For *best-arm identification*, we provide an $(\varepsilon, \delta)$-PAC algorithm with arm-memory size of $O(\log^* n)$ and $O(\frac{n}{\varepsilon^2} \cdot \log(\frac{1}{\delta}))$ optimal sample complexity.

## 1 Introduction

The Stochastic Multi-armed Bandit (MAB) problem is a classical framework used to capture decision-making in uncertain environments. In this model, a decision-maker is faced with $n$ choices (called *arms*) and has to sequentially choose one of the $n$ arms (referred to as *pulling* an arm). Based on the pulled arm, the decision-maker gets a reward drawn from a corresponding reward distribution unknown to the decision-maker. Starting with the seminal work of Robbins [26], MAB has been extensively studied with one of the following two goals: *regret minimization* and *best-arm identification*. See, e.g. [7] for a textbook treatment of the area. In addition to being theoretically interesting, MAB also finds numerous practical applications in diverse areas, including online advertising [31], crowd-sourcing [32], and clinical trials [9]. Hence, the study of MAB and its variants is of paramount interest in multiple fields, including online learning and reinforcement learning.

In many applications of the MAB problem, the number of arms (set of advertisements, crowd-workers, etc.) could be huge. Thus, the MAB problem with a large set of arms has received significant attention in recent years. Starting with the formative work of Berry et al. [6] and Herschkorn et al. [16] which studied the MAB problem with infinitely many arms, to the more recent work of Kleinberg et al. [20] which studies the MAB problem where the set of arms form a metric space, this variant has remained an active and prominent area of research. Large number of arms may enforce a constraint that the algorithm may not be able to store all the arms in the memory simultaneously. Additionally, as is common in these applications, the arms could arrive online, i.e., the algorithm may not have access to the entire set of arms at the beginning.

The streaming model, first formalized in the pioneering work of Alon et al. [3], has been developed to handle data streams where a huge amount of data arrives online and an algorithm has access

---

* A part of this work was done when the author was an undergraduate summer intern at IISc, Bangalore.

35th Conference on Neural Information Processing Systems (NeurIPS 2021).

to only a limited amount of memory. We refer the readers to [23, 2] for a detailed survey of streaming algorithms. With the ever-increasing amount of data available to any machine learning (ML) algorithm, several salient areas of ML are being studied under streaming constraints; e.g., active learning [35, 29], lifelong learning [33], and identifying correlations in multivariate data [11] (see the recent survey by Gomes et al. [15] for further discussions).

In this work, we study the classical stochastic MAB problem under streaming constraints. This model has recently been studied in [27, 21, 4]. In particular, we study a setting where we are given a set of $n$ arms arriving one-by-one, and an algorithm can store only a fixed number of arms $m < n$ in the memory. The set of arms stored in the memory is known as arm-memory and $m$ denotes the upper bound on the number of arms that can be stored in the memory at any time step. In this model, at any time step an arm not in the current arm-memory can not be pulled. Further, for any arm currently not in the arm-memory, we assume we do not know anything about the reward statistics corresponding to that arm until that arm is read into arm-memory and subsequently pulled. Note that the classical MAB algorithms such as UCB1 [5] and Thompson Sampling [30] for regret minimization, and Median-Elimination [13] for best-arm identification are not suited for this streaming setting as they require all arms to be present in the arm-memory at the start of the algorithm. Our goal is to study the interplay between *arm-memory size* vs. *expected regret* and *arm-memory size* vs. *sample complexity*, respectively.

An algorithm for our setting works as follows: at any time step $t$, the algorithm can only pull an arm that is currently in arm-memory and update its reward statistics. Then, the algorithm may choose to discard some of the arms that are currently in the arm-memory and read new arms into the arm-memory before proceeding to the next time-step to pull an arm from the arm-memory. In streaming terminology, algorithms that can not read back an arm into the arm-memory that was previously discarded from the arm-memory, are called *single-pass* or *one-pass* algorithms. On the other hand, algorithms that are allowed to read back an arm into the arm-memory that was previously discarded from the arm-memory, are called *multi-pass* algorithms. We note that our algorithms are designed for the more challenging single-pass setting, which has also been the focus of other prominent areas in machine learning, for example, Rai et al. [25] focus on designing one-pass SVMs when the data arrives in a stream, whereas Carvalho and Cohen [8] design single-pass algorithms for online learning.

## 1.1 Our Contribution

We study MAB under *bounded arm-memory*, where $n$ arms arrive in a stream and at most $m < n$ arms can be stored in the arm-memory at any time. We study the trade-off between arm-memory size vs. expected regret, and arm-memory size vs. sample complexity.

**Regret minimization:** Our first result, in Section 3, settles an open question stated in both [21] and [27] pertaining to the lower bound on the expected cumulative regret in this model. Using information-theoretic machinery related to KL-divergence, we show that any single-pass algorithm in this model will incur an expected regret of $\Omega\big(n^{1/3} \cdot T^{2/3}/m^{7/3}\big)$. Interestingly, this result holds for any $m < n$, which shows that even if the algorithm is allowed to store $n-1$ arms in the arm-memory at any time, we cannot hope to obtain a better regret guarantee in terms of $T$. This lower bound almost matches with the $\tilde{O}(n^{1/3}T^{2/3})$ bound on the expected cumulative regret, obtained by the standard uniform-exploration algorithm where we store $m = 2$ arms. We note here that the uniform-exploration algorithm with $m = 2$ can be trivially extended to an algorithm that stores $m > 2$ arms in the memory and achieves a regret guarantee that matches the lower bound by storing the 2 arms required by the uniform-exploration algorithm and filling the remaining $m - 2$ sized arm-memory with arbitrary arms that would have otherwise been discarded. Since this is already optimal for the single-pass setting, we cannot hope to get a better regret guarantee in terms of $T$, even if we store $n-1$ arms in the memory. This shows an interesting dichotomy between $m = n - 1$ and $m = n$, as one can obtain expected cumulative regret of $\tilde{O}\big(\sqrt{nT}\big)$ by standard UCB1 algorithm [5], where we are allowed to store $n$ arms. Our lower bound holds even when the arms arrive in a uniformly random sequence.

**Best-arm identification:** In Section 4, we propose an $(\varepsilon, \delta)$-PAC streaming algorithm that takes $r$ as input and stores $r + 1$ arms in the arm-memory at any time and has sample complexity $O(\frac{n}{\varepsilon^2} \cdot$

$(\mathtt{ilog}^{(r)}(n) + \ln(\frac{1}{\delta})))$ [2]. In particular, when $r = \log^* n$, our algorithm achieves the optimal worst-case sample complexity $O\left(\frac{n}{\varepsilon^2}\log(\frac{1}{\delta})\right)$ for any best-arm identification algorithm [13].

This problem was also studied in recent STOC'20 paper by Assadi and Wang [4] where they proposed an $(\varepsilon, \delta)$-PAC algorithm with optimal sample complexity and $O(\log^* n)$ arm-memory size. However, we show that due to an error in their analysis, their claim is incorrect and their algorithm is *not* $(\varepsilon, \delta)$-PAC. In Appendix E, we construct a family of input instances for which their algorithm will output a non-$\varepsilon$-best arm with probability significantly larger than $\delta$. In Section 4, we show that the special case of our algorithm (when $r = \log^* n$) indeed provides the guarantee claimed in [4].

## 1.2  Related Work

In one of the earliest works on MAB with bounded arm-memory, Herschkorn et al. [16] study the infinite-armed MAB problem where only a single arm is stored in the arm-memory at any time step and an arm that is once discarded from the arm-memory cannot be recalled. On the other hand, Cover [10] studied a different variant of bounded memory where the algorithm can remember the rewards of only a finite number of samples for each arm in a two-armed bandit model. The work of Lu and Lu [22] also studies this variant of bounded memory in the learning from experts setting, where the constraint is on the number of time steps for which the rewards can be stored by the algorithm and not on the number of arms.

More recently, Kalvit and Zeevi [18] studied the infinite-armed bandit problem where each arm is associated with a *type*, and the number of types is finite. An algorithm knows the set of types but not the exact type of each arm. For this problem, they propose an algorithm that samples a finite number of arms and has $O(\log T)$ expected cumulative regret. Further, de Heide et al. [12] also study the infinite-armed bandit problem where there could be multiple optimal arms. They propose an algorithm and prove that it achieves $O(\log T)$ expected regret by sampling $O(\log T)$ number of arms. In our work, we focus on the finite-armed MAB problem with bounded arm-memory. Recently, the MAB problem where arms arrive in a stream and arm-memory is bounded, has been studied in both regret minimization and best-arm identification frameworks.

For the regret minimization problem in the stochastic MAB setting, Peköz [24] showed that when $m = 1$, a single-pass algorithm can have linear regret unless the reward distributions of arms satisfy some additional conditions. Liau et al. [21] and RoyChaudhuri and Kalyanakrishnan [27] study the stochastic MAB problem with bounded arm-memory with the goal of minimizing the expected cumulative regret over $T$ time steps. The algorithms in both these works are multi-pass algorithms whereas our work focusses on single-pass algorithms. Liau et al. [21] propose an upper-confidence bound based algorithm with $O(1)$ arm-memory size, which achieves an expected cumulative regret bound of $O\left(\sum_{i \neq i^*} \log(\frac{\Delta_i}{\Delta})\frac{\log T}{\Delta_i}\right)$, which is within $\log(\frac{\Delta_i}{\Delta})$ factor of the UCB1 regret bound [5]. Here $\Delta$ is the gap parameter, i.e., the difference in the expected rewards of the best and the second-best arms, and $\Delta_i$ is the difference in the expected rewards of the best arm and the $i$'th arm. However, they do not provide a worst-case regret bound for their algorithm, and hence, depending on the instance the regret of this algorithm may be very high. More precisely, if say $\Delta_i$ is $1/T^{2/3}$, then the proven regret bound implies $O(T^{2/3})$ expected cumulative regret. RoyChaudhuri and Kalyanakrishnan [27] propose an algorithmic framework, which is given $m$ as input and uses a MAB algorithm as a black-box. When the MAB algorithm used is UCB1, their algorithm achieves an expected regret of $O(nm + n^{3/2}m\sqrt{T\log(T/nm)})$ using $\Omega(\log T)$ passes. The recent work of Xu and Zhao [34] also studies the regret minimization problem with bounded arm-memory constraints. However, they consider the setting with adversarial rewards and is different from the stochastic MAB setting studied in our paper.

Recently, Assadi and Wang [4] studied the best-arm identification variant of this problem. First, they propose an algorithm for the best-coin identification problem (equivalent to MAB with Bernoulli reward distributions), which keeps exactly one extra coin in the memory at any time and has optimal sample complexity. They further extend their algorithm to the top-$k$ coins identification problem, which stores $k$ coins in the memory and has optimal sample complexity. Crucially though, both algorithms assume that the gap parameter $\Delta$ is known. Throughout our work, we deal with the case when $\Delta$ is not known to the algorithm. Falahatgar et al. [14] also study the best-arm identification

---

[2]Here $\mathtt{ilog}^{(r)}$ is iterated logarithm of order $r$ (see Section 1.3).

problem with bounded arm-memory and provide an algorithm with $O(1)$ arm-memory size. However, they only provide a high probability guarantee on the sample complexity while additionally assuming that the arms arrive in a uniformly random sequence. Note that unlike [14], the sample complexity of our algorithm is independent of the rewards observed during the sampling, and we do not require the assumption that the arms need to arrive in a uniformly random sequence. In a recent independent work, Jin et al. [17] proposed an $\varepsilon$-top-$k$ arms identification algorithm with optimal sample complexity and arm-memory size of one. They also propose an algorithm that identifies the best-arm with high probability using arm-memory size equal to one, having optimal instance-dependent sample complexity, and using instance-dependent constant number of passes. We would like to note that their work provides high probability bounds on the sample complexity, whereas our work provides worst-case bounds on the same quantity.

## 1.3   Notation

Let $[k]$ (where $k \in \mathbb{N}$) denote the set $\{1, 2, \ldots, k\}$. Let $\log$ denote the binary logarithm. For integers $r \geq 0$, and $a \geq 1$, $\mathtt{ilog}^{(r)}(\cdot)$ denotes the iterated logarithm of order $r$, i.e., $\mathtt{ilog}^{(r)}(a) = \max\{\log(\mathtt{ilog}^{(r-1)}(a)), 1\}$ and $\mathtt{ilog}^{(0)}(a) = a$. Hence, $\mathtt{ilog}^{(\log^* n)}(n) = 1$. Let $\mathbb{P}, \mathbb{E}$ denote probability and expectation, respectively.

## 2   Model and Problem Definition

An instance of the MAB problem is defined as a tuple $\langle n, (\mathcal{D}_i)_{i \in [n]}, (\mu_i)_{i \in [n]} \rangle$ where $n$ is the number of arms. A pull of $\mathtt{arm}_i$ gives a reward in $[0, 1]$ drawn from a distribution $\mathcal{D}_i$ with mean $\mu_i \in [0, 1]$ that is unknown to the decision-maker beforehand. We study MAB under *bounded arm-memory* setting where the arms arrive in a stream, and at any time-step, the algorithm can store only $m < n$ arms in the arm-memory.

In a single-pass algorithm, every arm arrives only once and it can't be read back into the arm-memory after it is discarded from the arm-memory. When an arm is read into the arm-memory, its reward statistics get initialized to zero and are updated whenever the arm is pulled until it is discarded from the arm-memory. As we consider single-pass algorithms, our results hold even for a more general model where we do not enforce any restriction on the number of arms for which we can store the arm-statistics. We only impose a condition that at any time step the pulled arm can only be chosen from the set of $m$ arms present in the arm-memory. We cannot pull an arm if the arm is not present in the arm-memory. Next, we formalize the notions of regret minimization and best-arm identification.

The regret of a MAB algorithm can be thought of as the loss suffered by it due to not knowing the reward distributions of the arms beforehand. Let $i^* = \arg\max_{i \in [n]} \mu_i$. Then $\mathtt{arm}_{i^*}$ is the best arm and let $\mu^* = \mu_{i^*}$. The cumulative regret (also called the *pseudo-regret*) of an algorithm over $T$ time-steps is defined as follows:

**Definition 1** (Cumulative Regret). *Given an instance $\langle n, (\mathcal{D}_i)_{i \in [n]}, (\mu_i)_{i \in [n]} \rangle$ of the MAB problem, the cumulative regret of an algorithm after $T$ rounds is defined as $R(T) = \mu^* \cdot T - \sum_{t=1}^{T} \mu_{i_t}$, where $\mathtt{arm}_{i_t}$ is the arm pulled by the algorithm at time $t \in [T]$, and $\mu^* = \max_{i \in [n]} \mu_i$.*

The expected cumulative regret of an algorithm is defined as $\mathbb{E}[R(T)] = \mu^* \cdot T - \sum_{t=1}^{T} \mathbb{E}[\mu_{i_t}]$, where the expectation is over the randomness in the algorithm and the realization of rewards. In the model with bounded arm-memory, the goal is to minimize expected cumulative regret while storing at most $m \ (< n)$ arms in arm-memory at any time-step. Note that, popular algorithms such as UCB1 [5] and Thompson sampling [30] store all $n$ arms in memory, i.e., they require an arm-memory size equal to $n$.

For best-arm identification, the goal of a decision-maker is to output the best arm $\mathtt{arm}_{i^*}$ using the minimum number of arm pulls. In practice, a relaxed goal is to find an arm which is close to the best arm in terms of the expected reward. We formalize this notion below.

**Definition 2** ($\varepsilon$-best arm). *Given a parameter $\varepsilon \in (0, 1)$, $\mathtt{arm}_i$ with mean reward $\mu_i$ is said to be an $\varepsilon$-best arm if $\mu_i \geq \mu^* - \varepsilon$. Otherwise we call the arm $\mathtt{arm}_i$ to be a non-$\varepsilon$-best arm.*

**Definition 3** (($\varepsilon, \delta$)-PAC Algorithm). *Given an approximation parameter $\varepsilon \in [0, 1)$ and a confidence parameter $\delta \in [0, 1/2)$, an algorithm $\mathcal{A}$ is an ($\varepsilon, \delta$)-PAC algorithm if it outputs an $\varepsilon$-best arm with probability at least $1 - \delta$.*

Traditionally, the goal in the best-arm identification is to design an $(\varepsilon, \delta)$-PAC algorithm minimizing the total number of arm pulls. The goal under streaming setup is to find an $(\varepsilon, \delta)$-PAC algorithm that minimizes the total number of arm pulls while storing at most $m$ arms in the arm-memory at any time.

## 3 Regret Minimization

In this section, we study limitations of bounded arm-memory for regret minimization. An adaptation of uniform-exploration algorithm (see [28]) achieves expected cumulative regret of $\tilde{O}(n^{1/3}T^{2/3})$ with an arm-memory of two.[3] The algorithm keeps in the arm-memory one arm $\texttt{arm}^*$, called the king, with the best empirical mean $\widehat{\mu}_{\texttt{arm}^*}$ among the arms seen so far. Whenever a new arm $\texttt{arm}_i$ arrives, $\texttt{arm}_i$ is sampled $(T/n)^{2/3}O(\log T)^{1/3}$ times to obtain its empirical mean $\widehat{\mu}_i$. Then, $\widehat{\mu}_i$ is compared with $\widehat{\mu}_{\texttt{arm}^*}$. If $\widehat{\mu}_{\texttt{arm}^*} < \widehat{\mu}_i$, then $\texttt{arm}_i$ becomes the new king, replacing $\texttt{arm}^*$. Otherwise, $\texttt{arm}^*$ continues to be the king. After the algorithm tries out all the arms, it returns the king as the best-arm and continues to sample it for the rest of the time horizon.

An important question left open in [27] is to provide a lower bound on the expected cumulative regret of an algorithm with bounded arm-memory. We settle the question by showing that any single-pass algorithm for such a setting incurs at least $\Omega\left(n^{1/3} \cdot T^{2/3}/m^{7/3}\right)$ regret.

Now we state some fundamental properties of KL-divergence that will be needed in this section.

**Theorem 1** (Slivkins [28]). KL-*divergence satisfies the following properties:*

- Pinsker's inequality: *For any event $A \subset \Omega$, we have $2(p(A) - q(A))^2 \leq \texttt{KL}(p, q)$.*

- Chain rule for product distributions: *Let the sample space be a product $\Omega = \Omega_1 \times \Omega_2 \times \ldots \times \Omega_t$. Let $p, q$ be two distributions on $\Omega$ such that $p = p_1 \times p_2 \times \cdots \times p_t$ and $q = q_1 \times q_2 \times \cdots \times q_t$, where $p_j, q_j$ are distributions on $\Omega_j$, for each $j \in [t]$. Then $\texttt{KL}(p, q) = \sum_{j=1}^{t} \texttt{KL}(p_j, q_j)$.*

- Random coins: *Let $\texttt{B}_\varepsilon$ denote a Bernoulli distribution with mean $(1 + \varepsilon)/2$. Then $\texttt{KL}(\texttt{B}_\varepsilon, \texttt{B}_0) \leq 2\varepsilon^2$ and $\texttt{KL}(\texttt{B}_0, \texttt{B}_\varepsilon) \leq \varepsilon^2$, for all $\varepsilon \in (0, 1/2)$.*

For simplicity of presentation, we first discuss a weaker version of our main result in the following theorem.

**Theorem 2.** *In the* MAB *setting, fix the number of arms $n$ and the time horizon $T$. For any single-pass streaming MAB algorithm, if we are allowed to store at most $m < n$ arms, then there exists a problem instance such that $\mathbb{E}[R(T)] \geq \Omega(T^{2/3}/m^2)$*

*Proof.* We consider 0-1 rewards and the following family of problem instances $\{\mathcal{I}_j : j \in \{0, \ldots, m\}\}$ each containing $n$ arms, with parameter $\varepsilon > 0$ (where $\varepsilon = 1/T^{1/3}$):

$$\mathcal{I}_0 = \begin{cases} \mu_i & = 1/2, & \text{for } i \neq n; \\ \mu_i & = 1, & \text{for } i = n. \end{cases}$$

$$\forall j \in [m], \ \mathcal{I}_j = \begin{cases} \mu_i & = (1 + \varepsilon)/2, & \text{for } i = j; \\ \mu_i & = 1/2, & \text{for } i \neq j. \end{cases}$$

In the above instances, $\mu_i$ denotes the expected reward of $\texttt{arm}_i$, the $i$-th arm to arrive in the stream.

Before delving into the formal proof, we present an intuitive idea behind the proof. Consider the first $m$ arms that are loaded in the arm-memory. If these arms are not sampled for a sufficient number of times, then the probability of discarding the best-arm from the arm-memory is large enough to lead to a large regret in one of the instances in $\{(\mathcal{I}_j)_{j \in [m]}\}$. Instead, if these arms are sampled a greater number of times, we accumulate a large regret in the problem instance $\mathcal{I}_0$. Appropriately choosing the number of samplings to be done and balancing the regrets in these different cases leads to the lower bound on the regret.

Note that a deterministic algorithm does not lose anything by first loading the first $m$ arms into its memory. Thus, w.l.o.g. we consider a deterministic algorithm $\mathcal{A}$ which directly stores the first $m$ arms in the arm-memory.

---

[3]where $f(T) = \tilde{O}(g(T)) \implies f(T) = O(g(T) \log^k T)$ for some $k$.

We next set up the sample space. Let $L = 1/(4m^2\varepsilon^2)$. Further, let $(r_s(i) : i \in [m], s \in [L])$ be a table of mutually independent Bernoulli random variables where $r_s(i)$ has expectation $\mu_i$. We interpret $r_s(i)$ as the reward obtained when $\texttt{arm}_i$ is pulled for the $s$-th time and the table is called the *rewards table*. The sample space is then expressed as $\Omega = \{0, 1\}^{m \times L}$. Then, any $\omega \in \Omega$ can be interpreted as a realization of the rewards table.

Each instance $\mathcal{I}_j$, where $j \in \{0, \ldots, m\}$, defines a distribution $P_j$ on $\Omega$ as follows:

$$P_j(A) = \mathbb{P}[A \mid \mathcal{I}_j], \quad \text{for each } A \subseteq \Omega.$$

Given an instance $\mathcal{I}_j$ where $j \in \{0, \ldots, m\}$, let $P_j^{i,s}$ be the distribution of $r_s(i)$ under this instance. Then, we have that $P_j = \prod_{i \in [m], s \in [L]} P_j^{i,s}$.

Let $S_t \subseteq \{\texttt{arm}_1, \texttt{arm}_2, \ldots, \texttt{arm}_m\}$ denote the subset of first $m$ arms which are discarded from the arm-memory till (and including) time step $t$ by the algorithm $\mathcal{A}$. If $\texttt{arm}_i$ is discarded before the algorithm begins pulling arms, call this time step 0, then we include $\texttt{arm}_i$ in the set $S_0$ where $i \in [m]$. As time horizon $T$ is fixed , and we will eventually discard all arms at the end of time horizon, we can assume that $S_T = \{\texttt{arm}_1, \texttt{arm}_2, \ldots, \texttt{arm}_m\}$. For all $\omega \in \Omega$, let $T'_\omega = \arg\min_{0 \leq t \leq T}\{t : S_t \neq \emptyset\}$, i.e., $T'_\omega$ is the number of time steps since the beginning of the algorithm $\mathcal{A}$ when some arm in $\{\texttt{arm}_1, \texttt{arm}_2, \ldots, \texttt{arm}_m\}$ is discarded from the arm-memory for the first time. Let $A_1 = \{\omega \in \Omega : T'_\omega \leq L\}$ be the set of reward realizations for which $T'_\omega \leq L$. Now, fix some arm $i \in \{\texttt{arm}_1, \texttt{arm}_2, \ldots, \texttt{arm}_m\}$. Define $A_2^i = \{\omega \in \Omega : \texttt{arm}_i \in S_{T'_\omega}\}$ to be the event that the $\texttt{arm}_i$ belongs to $S_{T'_\omega}$. Now, let $A^i = A_1 \cap A_2^i$ be the set of reward realizations such that $\forall \omega \in A^i, T'_\omega \leq L$ and $\texttt{arm}_i$ is discarded from the arm-memory at the time step $T'_\omega$. Also, for any event $A \subseteq \Omega$, let $\overline{A} = \Omega \setminus A$.

Now we have the following observation for instance $\mathcal{I}_0$.

**Observation 1.** *If $\omega \in \overline{A_1}$, then the algorithm $\mathcal{A}$ would incur regret of at least $L/2 = T^{2/3}/(8m^2)$ on instance $\mathcal{I}_0$.*

*Proof.* The event $\overline{A_1}$ is the set of all reward realizations such that $T'_\omega$, the first time step at which some arm is discarded from the memory, is greater than $L$. Under instance $\mathcal{I}_0$, the first $m$ arms have mean $1/2$ whereas the best arm has mean 1. Hence, the regret in the first $L$ rounds is exactly $(1 - 1/2)L$. Substituting the value of $L$ gives us the observation. $\qquad\square$

Let $i' = \arg\max_{i \in [m]} P_0(A^i)$. We get the next observation as the best arm $\texttt{arm}_{i'}$ is sampled at most $L$ times.

**Observation 2.** *For all $\omega \in A^{i'}$, the regret on instance $\mathcal{I}_{i'}$ is at least $\varepsilon(T - L)/2 \geq T^{2/3}/4$.*

*Proof.* The event $A^{i'}$ is the set of all outcomes such that $T'_\omega$ , the first time step at which some arm is discarded from the memory, is less than $L$ and $\texttt{arm}_{i'}$ is one of the discarded arms. In instance $\mathcal{I}_{i'}$ , arm $i'$ has mean $(1 + \varepsilon)/2$ whereas other arms have mean $1/2$. Hence, discarding it within $L$ rounds leads to some sub-optimal arm being pulled in the remaining $T - L$ rounds. Thus, the regret in the last $T - L$ rounds is exactly $\varepsilon(T - L)/2$. Substituting the value of $L$ and $\varepsilon$ gives us the observation. $\qquad\square$

Now we will prove the following inequality, which will be useful in our analysis.

$$m \cdot P_{i'}(A^{i'}) + P_0(\overline{A_1}) \geq 1/4. \tag{1}$$

The above inequality intuitively shows that the probability of "bad" events that incur large regret, is significant. This inequality is trivially true if $P_0(\overline{A_1}) \geq 1/4$. Therefore, let us assume $P_0(\overline{A_1}) \leq 1/4$, i.e., $P_0(A_1) \geq 3/4$. Then $P_0(A^{i'}) \geq 3/(4m)$, by averaging argument. Using Theorem 1 for

distributions $P_0$ and $P_{i'}$, we obtain:

$$2(P_0(A^{i'}) - P_{i'}(A^{i'}))^2$$

$$\leq \mathtt{KL}(P_0, P_{i'}) = \sum_{i \in [m]} \sum_{t=1}^{L} \mathtt{KL}(P_0^{i,t}, P_{i'}^{i,t}) \qquad \text{(by Pinsker's inequality and chain rule)}$$

$$= \sum_{i \in [m] \setminus \{i'\}} \sum_{t=1}^{L} \mathtt{KL}(P_0^{i,t}, P_{i'}^{i,t}) + \sum_{t=1}^{L} \mathtt{KL}(P_0^{i',t}, P_{i'}^{i',t}) \leq 0 + L \cdot 2\varepsilon^2.$$

In the last inequality, the first term of the summation is zero because all arms $\mathtt{arm}_i$, where $i \in [m] \setminus \{i'\}$, have identical reward distributions under instances $\mathcal{I}_0$ and $\mathcal{I}_{i'}$. To bound the second term in the summation, we use the last property from Theorem 1. Thus, we have, $P_0(A^{i'}) - P_{i'}(A^{i'}) \leq \varepsilon\sqrt{L}$. Hence, $P_{i'}(A^{i'}) \geq P_0(A^{i'}) - \varepsilon\sqrt{L} \geq (3/(4m)) - (1/(2m)) = 1/(4m)$. Here, we use $P_0(A^{i'}) \geq 3/(4m)$ and $L = 1/(4m^2\varepsilon^2)$. Hence, if $P_0(\overline{A_1}) \leq 1/4$ then $m \cdot P_{i'}(A^{i'}) \geq 1/4$. This proves Inequality (1).

Now suppose that we choose an instance $\mathcal{I}$ in such a way that we have $\mathbb{P}[\mathcal{I} = \mathcal{I}_j] = 1/(2m)$ for any $j \in [m]$ and $\mathbb{P}[\mathcal{I} = \mathcal{I}_0] = 1/2$. Then,

$$\mathbb{E}[R(T)] \geq \mathbb{P}[\mathcal{I} = \mathcal{I}_{i'}] \cdot P_{i'}(A^{i'}) \cdot T^{2/3}/4 + \mathbb{P}[\mathcal{I} = \mathcal{I}_0] \cdot P_0(\overline{A_1}) \cdot T^{2/3}/(8m^2)$$

$$\geq \frac{1}{2m} \cdot (m \cdot P_{i'}(A^{i'})) \cdot T^{2/3}/(4m) + \frac{1}{2} \cdot P_0(\overline{A_1}) \cdot T^{2/3}/(8m^2)$$

$$\geq \left( m \cdot P_{i'}(A^{i'}) + P_0(\overline{A_1}) \right) \cdot T^{2/3}/(16m^2) \geq T^{2/3}/(64m^2)$$

The first inequality follows from Observations 1 and 2. In the last equality, we have used Inequality (1). Note that the expectation is taken over the choice of input instance and randomness in reward. $\quad\square$

Since a randomized algorithm is a distribution over deterministic algorithms, the above result also holds for any randomized algorithm. Now, by slight modification to the above family of instances and using more sophisticated techniques and involved analysis, we can improve the lower bound on the expected cumulative regret and obtain the following main result (see Appendix A for details):

**Theorem 3.** *In the* MAB *setting, fix the number of arms $n$ and the time horizon $T$. For any single-pass streaming* MAB *algorithm, if we are allowed to store at most $m < n$ arms, then there exists a problem instance such that $\mathbb{E}[R(T)] \geq \Omega(\frac{n^{1/3}T^{2/3}}{m^{7/3}})$.*

In the above result, the arms can arrive in any order, known as adversarial-order arrival. Our hardness results can also be extended to random-order arrival of arms, which we define next. Let $\langle n, (\mathcal{D}_i)_{i \in [n]}, (\mu_i)_{i \in [n]} \rangle$ be an instance of the MAB problem. Let, $\sigma : [n] \to [n]$ be a permutation and let $(\mathtt{arm}_{\sigma(i)})_{i \in [n]}$ be the ordering of $(\mathtt{arm}_i)_{i \in [n]}$ under $\sigma$. Define $\mathcal{S}_n = \{\sigma : \sigma$ is a permutation of $[n]\}$. Under the random-order arrival model, we assume that the arrival order of the arms in the stream is determined by a permutation $\sigma \in \mathcal{S}_n$, which is drawn uniformly at random from the set $\mathcal{S}_n$. The arms arrive in the order in which they appear in the tuple $(\mathtt{arm}_{\sigma(i)})_{i \in [n]}$, i.e., the first arm to arrive in the stream is $\mathtt{arm}_{\sigma(1)}$, followed by $\mathtt{arm}_{\sigma(2)}$, and so on. Random-order arrival is a well-studied model in optimization under uncertainty due to its connection with the secretary problem and optimal stopping theory [19]. The algorithm of RoyChaudhuri and Kalyanakrishnan [27] also uses an analogous random shuffling of arms. We show that even such random shuffling of arms does not beat $\Omega(T^{2/3})$ for the single-pass case. For random-order arrival, we obtain the following result (see Appendix B for details):

**Theorem 4.** *In the* MAB *setting, fix the number of arms $n$ and the time horizon $T$. For any single-pass streaming* MAB *algorithm, if we are allowed to store at most $m < n$ arms, then in the setting of random order arrival there exists an input instance such that $\mathbb{E}[R(T)] \geq \Omega(\frac{n^{1/3}T^{2/3}}{m^{7/3}})$.*

## 4 Best-arm Identification

In this section, we design an $(\varepsilon, \delta)$-PAC algorithm (Algorithm 1) which minimizes the total number of arm pulls while storing at most $m < n$ arms in the arm-memory. Let $r := m - 1$.

Intuitively, our algorithm maintains $r$ levels and in each level we maintain a running best-arm candidate (denoted as $\text{arm}_i^*$ for the arm in level $i$) in the arm-memory. Every new arm first arrives at the level 1. When an arm $\text{arm}'$ arrives at level $i$, we sample $\text{arm}'$ ($s_i$ number of times at level $i$) and compare its empirical mean with the empirical mean of $\text{arm}_i^*$. If it is better than $\text{arm}_i^*$ then $\text{arm}'$ is stored in $\text{arm}_i^*$ by replacing the previous arm in it, otherwise we drop $\text{arm}'$. Once we see a *sufficient* number ($= c_i$ for level $i$) of arms in a level, we send $\text{arm}_i^*$ to the next level $(i+1)$. At each level $i$, only one out of every $c_i$ arms is sent to level $i+1$. As the level index $i$ increases, value of $c_i$ and $s_i$ increases. Intuitively, for higher level indices, the number of arms reaching that level decreases rapidly. Hence, in higher levels each arm can be sampled more number of times for a more refined comparison without affecting the sample complexity. We perform a post-processing step at the end of the stream to send any arm present in intermediate levels to the final level for sampling. After the post-processing step, the algorithm returns the arm left in the highest level.

---

**Algorithm 1** $(\varepsilon, \delta)$-PAC algorithm for Best-arm indentification

---

1: $\{\varepsilon_\ell\}_{\ell=1}^r : \varepsilon_\ell \leftarrow \varepsilon/2^{\ell+1}$. //Intermediate gap parameter in level $\ell$.
2: $\{\beta_\ell\}_{\ell=1}^r : \beta_\ell \leftarrow 1/\varepsilon_\ell^2$.
3: $\{s_\ell\}_{\ell=1}^r : s_\ell \leftarrow \lceil 2\beta_\ell \big( \text{ilog}^{(r+1-\ell)}(n) + \log(\frac{2^{\ell+2}}{\delta}) \big) \rceil$. //Samples per arm in level $\ell$.
4: Stored arms: $\text{arm}', \text{arm}_1^*, \text{arm}_2^*, \ldots, \text{arm}_r^*$, where $\text{arm}_\ell^*$ is a running best-arm candidate at the $\ell$-th level, initialized to NULL.
5: Stored empirical means: $\mu_1^*, \mu_2^*, \ldots, \mu_r^*$, where $\mu_\ell^*$ is the highest empirical mean of $\ell$-th level, initialized to 0.
6: $\{c_\ell\}_{\ell=1}^r : c_\ell \leftarrow \lceil \text{ilog}^{(r-\ell)}(n) \rceil$. //Number of arms that need to arrive in level $\ell$ before we can promote $\text{arm}_\ell^*$ to level $\ell + 1$.
7: Counters: $C_1, C_2, \ldots, C_r$, initialized to 0.
   //Modified Selective Promotion (Steps 8-18).
8: **while** a new arm $\text{arm}_i$ arrives in the stream **do**
9:    $\text{arm}' \leftarrow \text{arm}_i$ and $\ell \leftarrow 0$.
10:   **while** $\ell = 0$ **or** $C_\ell = c_\ell$ **do**
11:      If $\ell \neq 0$, then $C_\ell \leftarrow 0$, $\mu_\ell^* \leftarrow 0$, $\text{arm}' \leftarrow \text{arm}_\ell^*$, and $\text{arm}_\ell^* \leftarrow \text{NULL}$.
12:      $\ell \leftarrow \ell + 1$. //Promoting $\text{arm}'$ to next level
13:      Sample $\text{arm}'$ for $s_\ell$ times and compare its empirical mean $\widehat{\mu}_{\text{arm}'}$ with $\mu_\ell^*$.
14:      If $\widehat{\mu}_{\text{arm}'} < \mu_\ell^*$, drop $\text{arm}'$ from the memory. Otherwise, $\text{arm}_\ell^* \leftarrow \text{arm}'$ and $\mu_\ell^* \leftarrow \widehat{\mu}_{\text{arm}'}$.
15:      $C_\ell \leftarrow C_\ell + 1$.
16:      If $C_\ell = c_\ell$ and $r = 1$, **return** $\text{arm}_1^*$ as the selected arm and terminate the Algorithm.
17:   **end while**
18: **end while**
19: If there is any arm which is stored in a level below $r$, then promote it to level $r$ and sample it for $s_r$ times. Let $\text{arm}_{\ell'}^*$ be the arm with the highest empirical mean $\widehat{\mu}_{\ell'}$ among the arms which were sampled in this step, where $\ell' \in [r-1]$. If $\mu_r^* \leq \widehat{\mu}_{\ell'}$ then $\text{arm}_r^* \leftarrow \text{arm}_{\ell'}^*$.
    //Post-processing step done at the end of the stream to send any arm present in intermediate levels to the final level for sampling.
20: **return** $\text{arm}_r^*$.

---

For $r = \log^* n$, our algorithm is related to the recent work by Assadi and Wang [4]. They propose an $(\varepsilon, \delta)$-PAC algorithm (Algorithm 3 in [4]) for the above setting, which has optimal sample complexity and stores at most $\log^* n$ arms in the arm-memory at any time step. However, their analysis is erroneous, due to which their algorithm will output a non-$\varepsilon$-best arm with probability much greater than $\delta$ for some input sequences. This implies that the $(\varepsilon, \delta)$-PAC guarantee in Lemma 6.2 of [4] does not hold. Here, we intuitively describe the problem in their analysis and show how our algorithm overcomes it.

The algorithm by [4] has a similar framework as that of our algorithm. It maintains $\log^* n$ levels and in each level they also maintain a running best-arm candidate (denoted as $\text{arm}_i^*$ for level $i$) in the arm-memory. However, their algorithm samples both $\text{arm}_\ell^*$ and the arriving arm $\text{arm}_i$ $s_\ell$ number of times whereas our algorithm only samples the arriving arm $\text{arm}_i$ $s_\ell$ number of times and reuses the stored empirical mean of $\text{arm}_\ell^*$ for the comparison. This seemingly simple but critical difference renders their claim erroneous. We briefly explain the intuition behind this with the help of an example. Let the first three arms to arrive in the stream be $\text{arm}_1, \text{arm}_2, \text{arm}_3$ with means

$0.5, 0.45, 0.40$, respectively. After sampling the first two arms, let us assume that $\texttt{arm}_2$ defeats $\texttt{arm}_1$, which can happen with sufficiently large probability as their means are close. Now, if we only sample $\texttt{arm}_3$ and use the previous empirical mean of $\texttt{arm}_2$, then $\texttt{arm}_3$ can defeat $\texttt{arm}_2$ if its empirical mean exceeds the empirical mean of $\texttt{arm}_2$ which in turn is greater than that of $\texttt{arm}_1$. This is unlikely as the means of $\texttt{arm}_3$ and $\texttt{arm}_1$ are far apart. On other hand, $\texttt{arm}_3$ has a good chance of defeating $\texttt{arm}_2$ if $\texttt{arm}_2$ is resampled since the empirical mean of $\texttt{arm}_2$ after resampling can be lower than the empirical mean of $\texttt{arm}_3$ with sufficiently large probability. Extending this example to a stream of $n$ arms would imply that an arm with significantly low mean can get selected by the Algorithm in [4] with high probability. For a detailed discussion on this, we refer the reader to Appendix E.

We now proceed towards the detailed analysis of Algorithm 1. For simplicity of presentation, we ignore the *ceiling* function in the expression of $s_\ell$ and $c_\ell$.

**Theorem 5.** *Algorithm 1 is an $(\varepsilon, \delta)$-PAC algorithm with worst-case sample complexity $O(\frac{n}{\varepsilon^2} \cdot (\texttt{ilog}^{(r)}(n) + \log(\frac{1}{\delta})))$ and $r + 1$ arm-memory size, where $1 \leq r \leq \log^*(n)$.*

*Proof.* We prove this theorem using the following lemmas. Lemma 7 shows that the sample complexity of our algorithm is $O(\frac{n}{\varepsilon^2} \cdot (\texttt{ilog}^{(r)}(n) + \ln(\frac{1}{\delta})))$. Lemma 9 gives the proof of correctness that the algorithm is $(\varepsilon, \delta)$-PAC. At any time step, we store the arm currently being sampled and furthermore, at each of the $r$ levels we store a single arm, implying the $r + 1$ arm-memory size of our algorithm. $\square$

**Corollary 6.** *Algorithm 1 is an $(\varepsilon, \delta)$-PAC algorithm with worst-case sample complexity $O(\frac{n}{\varepsilon^2} \cdot \log(\frac{1}{\delta}))$ and $O(\log^* n)$ arm-memory size, when $r = \log^* n$.*

The following lemma establishes the worst sample complexity of Algorithm 1 (see Appendix D for the detailed proof of the lemma).

**Lemma 7.** *The worst case sample complexity of the algorithm is $O(\frac{n}{\varepsilon^2} \cdot (\texttt{ilog}^{(r)}(n) + \ln(\frac{1}{\delta})))$ and it does not depend on the arms' rewards or gap parameter or the order in which the arms arrive.*

We use the following lemma which follows from the Hoeffding's inequality, to prove the correctness of our algorithm (see Appendix D for the detailed proof).

**Lemma 8.** *Let $\texttt{arm}_1$ and $\texttt{arm}_2$ be two different arms with means $\mu_1$ and $\mu_2$. Suppose $\mu_1 - \mu_2 \geq \theta$ and we sample each arm $\frac{K}{\theta^2}$ times to obtain empirical means $\widehat{\mu}_1$ and $\widehat{\mu}_2$. Then,*

$$\mathbb{P}(\widehat{\mu}_1 \leq \widehat{\mu}_2) \leq 2 \cdot e^{(-K/2)}$$

We now proceed to show that our algorithm is an $(\varepsilon, \delta)$-PAC algorithm. Towards that, we introduce the notion of reward gap. We say that an arm with mean $\mu_1$ has a reward gap of $\varepsilon'$ from an arm with mean $\mu_2$ if $\mu_2 - \mu_1 = \varepsilon'$. Now we prove the following claim.

**Claim 1.** *For any level $\ell < r$, let $\texttt{arm}'_\ell$ be the best arm to ever reach this level. Then, with probability at least $1 - \frac{\delta}{2^{\ell+1}}$, at least one arm with reward gap at most $\varepsilon_\ell$ from $\texttt{arm}'_\ell$ is sent to level $\ell + 1$ or is sampled in Step 19.*

*Proof.* Let us consider some level $\ell$ and a set of arms $\texttt{arm}_{\ell_1}, \texttt{arm}_{\ell_2}, \ldots, \texttt{arm}_{\ell c_\ell}$, which increases the counter $C_\ell$ from 0 to $c_\ell$. Denote the best arm among them as $\texttt{arm}'_\ell$ and let $\texttt{arm}^*_\ell$ be the empirically best arm seen so far since the arrival of $\texttt{arm}_{\ell_1}$. Our algorithm works as follows: once an arm arrives at level $\ell$, it is pulled $s_\ell$ number of times. Next, we compare its empirical mean with that of $\texttt{arm}^*_\ell$, which was computed when the arm corresponding to $\texttt{arm}^*_\ell$ arrived at level $\ell$. The arm with the greater empirical mean is maintained as $\texttt{arm}^*_\ell$ and its empirical mean is stored for future comparisons. Once $C_\ell = c_\ell$, $\texttt{arm}^*_\ell$ is sent to level $\ell + 1$. Note that this is equivalent to simultaneously pulling each of the $c_\ell$ arms $s_\ell$ number of times, and then sending the empirically best arm to level $\ell + 1$. Also note that if $0 < C_\ell < c_\ell$, and the condition in Step 8 is not satisfied then we will sample $\texttt{arm}^*_\ell$ in the Step 19.

Using Lemma 8, we can show that at any level $\ell$, if two arms have reward gap $\mu_i - \mu_j \geq \varepsilon_\ell$ then $\mathbb{P}[\widehat{\mu}_i < \widehat{\mu}_j] \leq \frac{\delta}{2^{\ell+1} \cdot c_\ell}$. Since at most $c_\ell$ arms arrive at level $\ell$ before the counter $C_\ell$ reaches $c_\ell$ or the condition in Step 8 is not satisfied, taking union bound we have that the probability that at least one arm with reward gap $\leq \varepsilon_\ell$ from $\texttt{arm}'_\ell$ is either sent to level $\ell + 1$ or is sampled in the Step 19 is at least $1 - \frac{\delta}{2^{\ell+1} \cdot c_\ell} \cdot c_\ell = 1 - \frac{\delta}{2^{\ell+1}}$. Note that while using the union bound, we are comparing the

empirical mean of each of the arms with the empirical mean of the best arm. So the number of events is at most $c_\ell$. □

**Claim 2.** *Let* $\texttt{arm}'_r$ *be the best arm among the arms which reached level $r$ including the arms which were sampled in Step 19 of Algorithm 1. Then, with probability at least $1 - \frac{\delta}{2^{r+1}}$, Algorithm 1 returns an arm with reward gap at most $\varepsilon_r$ from* $\texttt{arm}'_r$.

*Proof.* Using Lemma 8, we can show that among the arms that are being considered here, if two arms have reward gap $\mu_i - \mu_j \geq \varepsilon_r$ then $\mathbb{P}[\widehat{\mu}_i < \widehat{\mu}_j] \leq \frac{\delta}{2^{r+1} \cdot c_r}$. As $c_r = n$, by taking union bound we get that with probability at least $1 - \frac{\delta}{2^{r+1}}$, Algorithm 1 returns an arm with reward gap at most $\varepsilon_r$ from $\texttt{arm}'_r$. □

**Lemma 9.** *With probability at least $1 - \delta$, the arm selected by the algorithm is an $\varepsilon$-best arm.*

*Proof.* Let the best arm be $\texttt{arm}^*$. By union bound and Claim 1, the probability that no arm with reward gap at most $\sum_{i=1}^{r-1} \varepsilon_i$ from $\texttt{arm}^*$ reaches level $r$ is upper bounded by $\sum_{\ell=1}^{r-1} \delta/2^{\ell+1}$. Given this upper bound and Claim 2, the probability that Algorithm 1 doesn't return an arm with reward gap at most $\sum_{i=1}^{r} \varepsilon_i$ from $\texttt{arm}^*$ is upper bounded by: $\sum_{\ell=1}^{r} \delta/2^{\ell+1} \leq \delta \cdot \sum_{\ell=1}^{\infty} 1/2^{\ell+1} = \delta/2 < \delta$.

Thus with probability at least $1 - \delta$, Algorithm 1 returns an arm with reward gap at most: $\sum_{\ell=1}^{r} \varepsilon_\ell = \sum_{\ell=1}^{r} \varepsilon/2^{\ell+1} \leq \varepsilon/2 < \varepsilon$ from $\texttt{arm}^*$. This concludes the proof. □

## 5 Conclusion

In this paper, we studied the stochastic MAB problem with bounded arm-memory under regret minimization and best-arm identification. First we showed a lower bound of $\Omega\left(n^{1/3} \cdot T^{2/3}/m^{7/3}\right)$ on the expected cumulative regret for single-pass MAB algorithms with bounded arm-memory. Next we proposed $(\varepsilon, \delta)$-PAC streaming algorithm with $r + 1$ arm-memory size ($r \in [\log^* n]$) which has optimal sample complexity when $r = \log^* n$. Our best-arm identification algorithm is a streaming version of the $r$-round adaptive algorithm where the arm pulls in each round are decided based on the observed outcomes in the earlier rounds [1], and the best-arm is the output at the end of $r$ rounds. The upper bound on the sample complexity of our algorithm almost matches with the lower bound for any $r$-round adaptive algorithm that is mentioned in [1].

A future research direction is to obtain lower bounds and upper bounds on the expected cumulative regret for $k$-pass stochastic MAB algorithms with bounded arm-memory where $k > 1$. Another direction would be to obtain instance-dependent lower bound and upper bound on the expected cumulative regret for single-pass stochastic MAB algorithms with bounded arm-memory. Another interesting question is to obtain an $(\varepsilon, \delta)$-PAC streaming algorithm with $O(1)$ arm-memory size and optimal worst case sample complexity.

## Acknowledgements

Vishakha Patil gratefully acknowledges the support of a Google PhD Fellowship. Arindam Khan is thankful to be supported by a Pratiksha Trust Young Investigator Award. The authors would like to thank Santanu Rathod for useful discussions surrounding interesting problems in this area. The authors are also grateful to the anonymous reviewers for their detailed comments that helped us to improve the presentation of this work.

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
