# A Improved lower bound for regret minimization

**Theorem 10.** *In the* MAB *setting, fix the number of arms $n$ and the time horizon $T$. For any single-pass streaming* MAB *algorithm, if we are allowed to store at most $m < n$ arms, then there exists a problem instance such that $\mathbb{E}[R(T)] \geq \Omega(\frac{n^{1/3}T^{2/3}}{m^{7/3}})$.*

*Proof.* We consider 0-1 rewards and the following family of problem instances $\{\mathcal{I}_j : j \in \{0, \dots, n-1\}\}$ each containing $n$ arms, with parameter $\varepsilon > 0$ (where $\varepsilon = \frac{n^{1/3}}{m^{1/3}T^{1/3}}$):

$$\mathcal{I}_0 = \begin{cases} \mu_i & = 1/2, & \text{for } i \neq n; \\ \mu_i & = 1, & \text{for } i = n. \end{cases}$$

$$\forall j \in [n-1], \mathcal{I}_j = \begin{cases} \mu_i & = (1+\varepsilon)/2, & \text{for } i = j; \\ \mu_i & = 1/2, & \text{for } i \neq j. \end{cases}$$

In the above instances, $\mu_i$ denotes the expected reward of $\texttt{arm}_i$, the $i$-th arm to arrive in the stream. We choose the problem instance $\mathcal{I}_0$ with probability $1/2$ and choose the rest of the problem instances with probability $1/(2(n-1))$.

We now fix a deterministic algorithm $\mathcal{A}$. First, note that any randomized algorithm is a distribution over deterministic algorithms and hence a lower bound for deterministic algorithms also implies a lower bound for randomized algorithms. Further, we assume that at its termination after $T$ rounds, algorithm $\mathcal{A}$ has read every arm into the arm-memory at some time step less than or equal to $T$. Note that the arm may be discarded immediately without being pulled. Also, all arms are discarded from the memory after $T$ rounds. We note that this assumption does not restrict the class of algorithms for which our lower bound holds, since any algorithm that processes the arms differently can be replicated by an algorithm with the above assumption.

Let the regret incurred by $\mathcal{A}$ be denoted by $R(T)$. Then, we can write $R(T)$ as:

$$R(T) = \sum_{i=0}^{n-1} X_{\mathcal{I}_i}$$

where $X_{\mathcal{I}_i}$ denotes the regret incurred due to the problem instance $\mathcal{I}_i$. If the problem instance $\mathcal{I}_i$ is not an input to $\mathcal{A}$, then $X_{\mathcal{I}_i} = 0$. Now, we can represent $X_{\mathcal{I}_0}$ as the following:

$$X_{\mathcal{I}_0} = \sum_{i=1}^{n-1} X_i$$

where $X_i$ denotes the regret incurred by $\mathcal{A}$ due to sampling $\texttt{arm}_i$ of the problem instance $\mathcal{I}_0$. Note that, $X_i = 0$ if $\texttt{arm}_i$ was never sampled.

For each $k \in \{0, \dots, \lfloor \frac{n-1}{m} \rfloor - 1\}$, let $Y_k = \sum_{i=1}^{m} (X_{km+i} + X_{\mathcal{I}_{km+i}})$. Let $\ell = \lfloor \frac{n-1}{m} \rfloor - 1$. Note that,

$$\mathbb{E}[R(T)] \geq \sum_{k=0}^{\ell} \mathbb{E}[Y_k].$$

We now prove a lower bound on $\mathbb{E}[Y_k]$ which will give us a lower bound on the regret as desired.

Intuitively, $\mathbb{E}[Y_k]$ denotes the expected regret an algorithm would incur by sampling the arms in the instances $\mathcal{I}_{km+1}, \dots, \mathcal{I}_{km+m}$ and the arms $\texttt{arm}_{km+1}, \dots, \texttt{arm}_{km+m}$ in the instance $\mathcal{I}_0$. The reason for clubbing these arms together is that if the arms $\texttt{arm}_{km+1}, \dots, \texttt{arm}_{km+m}$ are sampled a large number of times, then we incur a huge regret in expectation in the instance $\mathcal{I}_0$ and otherwise if the arms $\texttt{arm}_{km+1}, \dots, \texttt{arm}_{km+m}$ are sampled only for a few times, then we risk sampling the best arm very few times in one of the instances $\mathcal{I}_{km+1}, \dots, \mathcal{I}_{km+m}$ which will eventually lead to a huge regret.

Let $\mathcal{R} = \{0, 1\}^{km \times T}$ be the set of all possible reward realizations of the first $km$ arms. Let us fix a reward realization $R' \in \mathcal{R}$ of the first $km$ arms. We next set up the sample space. Let $L = 1/(4m^2\varepsilon^2)$.

Let $(r_s(i) : i \in [m], s \in [L])$ be a table of mutually independent Bernoulli random variables where $r_s(i)$ has expectation $\mu_{km+i}$. We interpret $r_s(i)$ as the reward obtained when $\texttt{arm}_{km+i}$ is pulled for the $s$-th time and the table is called the *rewards table*. The sample space is then expressed as $\Omega = \{0,1\}^{m \times L}$. Then, any $\omega \in \Omega$ can be interpreted as a realization of the rewards table.

For a fixed $R'$, each instance $\mathcal{I}_{km+j}$, where $j \in [m]$, defines a distribution $P_{j,R'}$ on $\Omega$ as follows:
$$P_{j,R'}(A) = \mathbb{P}[A \mid \mathcal{I}_{km+j}, R'], \quad \text{for each } A \subseteq \Omega$$

Similarly, the instance $\mathcal{I}_0$ defines a distribution $P_0$ on $\Omega$ as follows:
$$P_{0,R'}(A) = \mathbb{P}[A \mid \mathcal{I}_0, R'], \quad \text{for each } A \subseteq \Omega$$

Given an instance $\mathcal{I}_{km+j}$ where $j \in [m]$, let $P^{i,s}_{j,R'}$ be the distribution of $r_s(i)$ under this instance. Then we have that $P_{j,R'} = \prod_{i \in [m], s \in [L]} P^{i,s}_{j,R'}$. Similarly given the instance $\mathcal{I}_0$, let $P^{i,s}_{0,R'}$ be the distribution of $r_s(i)$ under this instance. Then we have that $P_{0,R'} = \prod_{i \in [m], s \in [L]} P^{i,s}_{0,R'}$.

Let $Z^i_k$ be a running counter for the number of arm pulls of $\texttt{arm}_{km+i}$ that gets incremented by one everytime the arm is pulled. Let $Z_k = \sum_{i=1}^m Z^i_k$. Let $S_t \subseteq \{\texttt{arm}_{km+1}, \texttt{arm}_{km+2}, \ldots, \texttt{arm}_{km+m}\}$ denote the subset of arms which are discarded from memory by the algorithm $\mathcal{A}$ before $Z_k$ becomes $t+1$. As time horizon $T$ is fixed and we will eventually discard all arms at the end of time horizon, we can assume that there exists a $t' \in [T] \cup \{0\}$ such that $S_{t'} = \{\texttt{arm}_{km+1}, \texttt{arm}_{km+2}, \ldots, \texttt{arm}_{km+m}\}$. For all $\omega \in \Omega$, let $T'_\omega = \arg\min_{0 \le t \le T}\{t : S_t \neq \emptyset\}$, i.e., $T'_\omega$ is the minimum value of $Z_k$ when some arm in $\{\texttt{arm}_{km+1}, \texttt{arm}_{km+2}, \ldots, \texttt{arm}_{km+m}\}$ is discarded from memory for the first time. Let $A_1 = \{\omega \in \Omega : T'_\omega \le L\}$ be the set of reward realizations for which $T'_\omega \le L$. Now fix some arm in $\{\texttt{arm}_{km+1}, \texttt{arm}_{km+2}, \ldots, \texttt{arm}_{km+m}\}$ say $\texttt{arm}_{km+i}$. Define $A^i_2 = \{\omega \in \Omega : \texttt{arm}_{km+i} \in S_{T'_\omega}\}$ to be the event that the $\texttt{arm}_{km+i}$ belongs to $S_{T'_\omega}$. Now, let $A^i = A_1 \cap A^i_2$ be the set of reward realizations such that $\forall \omega \in A^i$, $T'_\omega \le L$ and $\texttt{arm}_i$ is discarded from memory before $Z_k$ becomes $T'_\omega + 1$. Also, for any event $A \subseteq \Omega$, let $\overline{A} = \Omega \setminus A$.

Now we have the following observation for instance $\mathcal{I}_0$.

**Observation 3.** *If $\omega \in \overline{A_1}$ and $\mathcal{I}_0$ is an input instance to the algorithm $\mathcal{A}$, then $\sum_{i=1}^m X_{km+i} \ge \Omega(\frac{1}{m^2 \varepsilon^2})$.*

Let $i' = \arg\max_{i \in [m]} P_{0,R'}(A^i)$. We obtain the next observation due to the fact that the best arm $\texttt{arm}_{km+i'}$ is sampled for at most $L = o(T)$ times.

**Observation 4.** *If $\omega \in A^{i'}$ and $\mathcal{I}_{km+i'}$ is an input instance to the algorithm $\mathcal{A}$, then $X_{\mathcal{I}_{km+i'}} \ge \frac{\varepsilon(T-T'_\omega)}{2} = \Omega(\varepsilon T)$.*

Now we will prove the following inequality which will be useful in our analysis:

$$m \cdot P_{i',R'}(A^{i'}) + P_{0,R'}(\overline{A_1}) \ge \frac{1}{4}. \tag{2}$$

The above inequality is trivially true if $P_{0,R'}(\overline{A_1}) \ge 1/4$. Therefore, let us assume $P_{0,R'}(\overline{A_1}) \le 1/4$, i.e., $P_{0,R'}(A_1) \ge 3/4$. Then $P_{0,R'}(A^{i'}) \ge 3/(4m)$, by averaging argument and the definition of $i'$. Using Theorem 1 for distributions $P_{0,R'}$ and $P_{i',R'}$, we obtain:

$$
\begin{aligned}
2(P_{0,R'}&(A^{i'}) - P_{i',R'}(A^{i'}))^2 \\
&\le \mathrm{KL}(P_{0,R'}, P_{i',R'}) && \text{(by Pinsker's inequality)} \\
&= \sum_{i \in [m]} \sum_{t=1}^L \mathrm{KL}(P^{i,t}_{0,R'}, P^{i,t}_{i',R'}) && \text{(by chain rule)} \\
&= \sum_{i \in [m]\setminus\{i'\}} \sum_{t=1}^L \mathrm{KL}(P^{i,t}_{0,R'}, P^{i,t}_{i',R'}) + \sum_{t=1}^L \mathrm{KL}(P^{i',t}_{0,R'}, P^{i',t}_{i',R'}) \\
&\le 0 + L \cdot 2\varepsilon^2.
\end{aligned}
$$

In the last inequality, the first term of the summation is zero because all arms $\texttt{arm}_{km+i}$, where $i \in [m] \setminus \{i'\}$, have identical reward distributions under instances $\mathcal{I}_0$ and $\mathcal{I}_{km+i'}$. To bound the second term in the summation, we use the last property from Theorem 1. Thus we have, $P_{0,R'}(A^{i'}) - P_{i',R'}(A^{i'}) \leq \varepsilon\sqrt{L}$. Hence, $P_{i',R'}(A^{i'}) \geq P_{0,R'}(A^{i'}) - \varepsilon\sqrt{L} \geq (3/(4m)) - (1/(2m)) = 1/(4m)$. Here, we use $P_{0,R'}(A^{i'}) \geq 3/(4m)$ and $L = 1/(4m^2\varepsilon^2)$. Hence, if $P_{0,R'}(\overline{A_1}) \leq \frac{1}{4}$ then $m \cdot P_{i',R'}(A^{i'}) \geq \frac{1}{4}$. This proves Inequality (2). Let $\mathcal{I}'$ be the input instance. Now we have the following:

$$\mathbb{E}[Y_k|R']$$

$$\geq \mathbb{P}[\mathcal{I}' = \mathcal{I}_{km+i'}|R'] \cdot P_{i',R'}(A^{i'}) \cdot \Omega(\varepsilon T)$$

$$+ \mathbb{P}[\mathcal{I}' = \mathcal{I}_0|R'] \cdot P_{0,R'}(\overline{A_1}) \cdot \Omega\left(\frac{1}{m^2\varepsilon^2}\right)$$

$$= \frac{\mathbb{P}[R'|\mathcal{I}' = \mathcal{I}_{km+i'}] \cdot \mathbb{P}[\mathcal{I}' = \mathcal{I}_{km+i'}]}{\mathbb{P}[R']} \cdot P_{i',R'}(A^{i'}) \cdot \Omega(\varepsilon T)$$

$$+ \frac{\mathbb{P}[R'|\mathcal{I}' = \mathcal{I}_0] \cdot \mathbb{P}[\mathcal{I}' = \mathcal{I}_0]}{\mathbb{P}[R']} \cdot P_{0,R'}(\overline{A_1}) \cdot \Omega\left(\frac{1}{m^2\varepsilon^2}\right)$$

(Due to Bayes' Theorem )

$$= \frac{\mathbb{P}[R'|\mathcal{I}' = \mathcal{I}_{km+i'}]}{\mathbb{P}[R']} \cdot \frac{1}{2(n-1)} \cdot m P_{i',R'}(A^{i'}) \cdot \Omega(\varepsilon T/m)$$

$$+ \frac{\mathbb{P}[R'|\mathcal{I}' = \mathcal{I}_0]}{\mathbb{P}[R']} \cdot P_{0,R'}(\overline{A_1}) \cdot \frac{1}{2} \cdot \Omega\left(\frac{1}{m^2\varepsilon^2}\right)$$

$$\geq \frac{\mathbb{P}[R'|\mathcal{I}' = \mathcal{I}_{km+i'}]}{\mathbb{P}[R']} \cdot \Omega\left(\frac{T^{2/3}}{m^{4/3}n^{2/3}}\right)$$

$$\cdot (m \cdot P_{i',R'}(A^{i'}) + P_{0,R'}(\overline{A_1}))$$

(Due to the fact that $\mathbb{P}[R'|\mathcal{I}' = \mathcal{I}_{km+i'}] = \mathbb{P}[R'|\mathcal{I}' = \mathcal{I}_0]$

and $\Omega\left(\frac{1}{m^2\varepsilon^2}\right) = \Omega\left(\frac{\varepsilon T}{mn}\right) = \Omega\left(\frac{T^{2/3}}{m^{4/3}n^{2/3}}\right)$)

$$\geq \frac{\mathbb{P}[R'|\mathcal{I}' = \mathcal{I}_{km+i'}]}{\mathbb{P}[R']} \cdot \Omega\left(\frac{T^{2/3}}{m^{4/3}n^{2/3}}\right)$$

(Due to Inequality (2)).

Therefore we now have the following:

$$\mathbb{E}[Y_k] \geq \sum_{R' \in \mathcal{R}} \mathbb{P}[R'] \cdot \mathbb{E}[Y_k|R']$$

$$\geq \sum_{R' \in \mathcal{R}} \mathbb{P}[R'|\mathcal{I}' = \mathcal{I}_{km+i'}] \cdot \Omega\left(\frac{T^{2/3}}{m^{4/3}n^{2/3}}\right)$$

$$\geq \Omega\left(\frac{T^{2/3}}{m^{4/3}n^{2/3}}\right).$$

Let $\ell = \lfloor \frac{n-1}{m} \rfloor - 1$. Then we have the following. :

$$\mathbb{E}[R(T)] \geq \sum_{k=0}^{\ell} \mathbb{E}[Y_k]$$

$$\geq \sum_{i=0}^{\ell} \Omega\left(\frac{T^{2/3}}{m^{4/3}n^{2/3}}\right)$$

$$\geq \Omega\left(\frac{n^{1/3}T^{2/3}}{m^{7/3}}\right).$$

$\square$

## B   Regret Minimization under Random Order Arrival

**Theorem 11.** *In the* MAB *setting, fix the number of arms $n$ and the time horizon $T$. For any one-pass streaming* MAB *algorithm, if we are allowed to store at most $m < n$ arms, then in the setting of random order arrival there exists an input instance such that $\mathbb{E}[R(T)] \geq \Omega(\frac{n^{1/3}T^{2/3}}{m^{7/3}})$.*

*Proof.* We consider 0-1 rewards and the 2 input instances $\mathcal{I}_1, \mathcal{I}_2$ each containing $n$ arms, with parameter $\varepsilon > 0$ (where $\varepsilon = \frac{n^{1/3}}{m^{1/3}T^{1/3}}$):

$$\mathcal{I}_1 = \begin{cases} \mu_i & = (1 + \varepsilon)/2 & \text{for } i = 1 \\ \mu_i & = 1/2, & \text{for } i \neq 1 \end{cases}$$

$$\mathcal{I}_2 = \begin{cases} \mu_i & = 1/2 & \text{for } i \neq n \\ \mu_i & = 1. & \text{for } i = n \end{cases}$$

In the above instances, $\mu_i$ denotes the expected reward of the $i^{th}$ arm in the input instance.
We now fix a deterministic algorithm $\mathcal{A}$. First, note that any randomized algorithm is a distribution over deterministic algorithms and hence a lower bound for deterministic algorithms also implies a lower bound for randomized algorithms. Further, we assume that at its termination after $T$ rounds, algorithm $\mathcal{A}$ has read every arm into the arm-memory at some time step less than or equal to $T$. Note that the arm may be discarded immediately without being pulled. Also, all arms are discarded from the memory after $T$ rounds. We note that this assumption does not restrict the class of algorithms for which our lower bound holds, since any algorithm that processes the arms differently can be replicated by an algorithm with the above assumption.

We choose an input instance uniformly at random from $\mathcal{I}_1$ and $\mathcal{I}_2$. Let this input instance be $\mathcal{I}'$. Then under random-order arrival setting one of the $n$ permutations of $\mathcal{I}'$ is chosen uniformly at random and is sent as the input stream to the Algorithm $\mathcal{A}$. Note that this is equivalent to choosing a permutation $\mathcal{P}$ from $2n$ total distinct permutations of $\mathcal{I}_1$ and $\mathcal{I}_2$ uniformly at random and sending it to the algorithm $\mathcal{A}$. Let $\mathcal{I}'_1$ be the collection of distinct permutations of $\mathcal{I}_1$ such that the arm with expected reward of $(1 + \varepsilon)/2$ is in the first $n/2$ positions of the permutation. Similarly, let $\mathcal{I}'_2$ be the collection of distinct permutations of $\mathcal{I}_2$ such that the arm with expected reward of $1$ is not in the first $n/2$ positions of the permutation. Clearly, $|\mathcal{I}'_1| = n/2$ and $|\mathcal{I}'_2| = n/2$. Let $\mathcal{I}^j_1$ denote a input permutation in $\mathcal{I}'_1$ such that the $\texttt{arm}_j$ has $\mu_j = (1 + \varepsilon)/2$.

Let the regret incurred by $\mathcal{A}$ be denoted by $R(T)$. Using arguments similar to the previous section, we will show that $\mathbb{E}[R(T)] \geq \Omega(\frac{n^{1/3}T^{2/3}}{m^{7/3}})$. First, note that we have the following:

$$R(T) \geq \sum_{i=1}^{n/2} X_{\mathcal{I}^j_1} + X_{\mathcal{I}'_2}$$

where $X_{\mathcal{I}^j_1}$ denotes the regret incurred due to the permutation $\mathcal{I}^j_1$ and $X_{\mathcal{I}'_2}$ denotes the total regret incurred due to set of permutations $\mathcal{I}'_2$. If the permutation $\mathcal{I}^j_1$ is not an input to $\mathcal{A}$, then $X_{\mathcal{I}^i_1} = 0$. Similarly if none of the permutations from $\mathcal{I}'_2$ is an input to to $\mathcal{A}$, then $X_{\mathcal{I}'_2} = 0$. Now we can represent $X_{\mathcal{I}'_2}$ as the following:

$$X_{\mathcal{I}'_2} = \sum_{i=1}^{n} X_i$$

where $X_i$ denotes the regret incurred by $\mathcal{A}$ due to sampling $\texttt{arm}_i$ of the set of permutations $\mathcal{I}'_2$. Note that $X_i = 0$ if $\texttt{arm}_i$ of the set of permutations $\mathcal{I}'_2$ were never sampled.

For each $k \in \{0, \ldots, \lfloor\frac{n}{2m}\rfloor - 1\}$, let $Y_k = \sum_{i=1}^{m}(X_{km+i} + X_{\mathcal{I}^{km+i}_1})$. We now show a lower bound on $\mathbb{E}[Y_k]$.

Intuitively, $\mathbb{E}[Y_k]$ denotes the expected regret an algorithm would incur by the sampling the arms in the permutations $\mathcal{I}^{km+1}_1, \ldots, \mathcal{I}^{km+m}_1$ and the arms $\texttt{arm}_{km+1}, \ldots, \texttt{arm}_{km+m}$ in the set of permutations $\mathcal{I}'_2$. The reason for clubbing these arms together is that if the arms $\texttt{arm}_{km+1}, \ldots, \texttt{arm}_{km+m}$ are

sampled for a large number of times, then we incur a huge regret in expectation in the set of permutations $\mathcal{I}'_2$ and otherwise if the arms $\texttt{arm}_{km+1}, \ldots, \texttt{arm}_{km+m}$ are sampled only a few times, then we risk sampling the best arm very few times in one of the permutations $\mathcal{I}_1^{km+1}, \ldots, \mathcal{I}_1^{km+m}$ which will eventually lead to a huge regret.

Let $\mathcal{R} = \{0,1\}^{km \times T}$ be the set of all possible reward realizations of the first $km$ arms. Let us fix a reward realization $R' \in \mathcal{R}$ of the first $km$ arms. We next set up the sample space. Let $L = 1/(4m^2\varepsilon^2)$. Let $(r_s(i) : i \in [m], s \in [L])$ be a table of mutually independent Bernoulli random variables where $r_s(i)$ has expectation $\mu_{km+i}$. We interpret $r_s(i)$ as the reward obtained when $\texttt{arm}_{km+i}$ is pulled for the $s$-th time and the table is called the *rewards table*. The sample space is then expressed as $\Omega = \{0,1\}^{m \times L}$. Then any $\omega \in \Omega$ can be interpreted as a realization of the rewards table.

Each permutation $\mathcal{I}_1^{km+j}$, where $j \in [m]$, defines a distribution $P_{j,R'}$ on $\Omega$ as follows:

$$P_{j,R'}(A) = \mathbb{P}[A \mid \mathcal{I}_1^{km+j}, R'], \quad \text{for each } A \subseteq \Omega$$

Let $\mathcal{P}$ be an input permutation. Then we define a distribution $P_0$ on $\Omega$ as follows:

$$P_{0,R'}(A) = \mathbb{P}[A \mid \mathcal{P} \in \mathcal{I}'_2, R'], \quad \text{for each } A \subseteq \Omega$$

Given a permutation $\mathcal{I}_{km+j}$ where $j \in [m]$, let $P_{j,R'}^{i,s}$ be the distribution of $r_s(i)$ under this permutation. Then we have that $P_{j,R'} = \prod_{i \in [m], s \in [L]} P_{j,R'}^{i,s}$. Similarly given an input permutation $\mathcal{P} \in \mathcal{I}'_2$, let $P_{0,R'}^{i,s}$ be the distribution of $r_s(i)$ under this permutation. Then we have that $P_{0,R'} = \prod_{i \in [m], s \in [L]} P_{0,R'}^{i,s}$.

Let $Z_k^i$ be a running counter for the number of arm pulls of $\texttt{arm}_{km+i}$ that gets incremented by one everytime the arm is pulled. Let $Z_k = \sum_{i=1}^m Z_k^i$. Let $S_t \subseteq \{\texttt{arm}_{km+1}, \texttt{arm}_{km+2}, \ldots, \texttt{arm}_{km+m}\}$ denote the subset of arms which are discarded from memory by the algorithm $\mathcal{A}$ before $Z_k$ becomes $t+1$. As time horizon $T$ is fixed and we will eventually discard all arms at the end of time horizon, we can assume that there exists a $t' \in [T] \cup \{0\}$ such that $S_{t'} = \{\texttt{arm}_{km+1}, \texttt{arm}_{km+2}, \ldots, \texttt{arm}_{km+m}\}$. For all $\omega \in \Omega$, let $T'_\omega = \arg\min_{0 \le t \le T}\{t : S_t \ne \emptyset\}$, i.e., $T'_\omega$ is the minimum value of $Z_k$ when some arm in $\{\texttt{arm}_{km+1}, \texttt{arm}_{km+2}, \ldots, \texttt{arm}_{km+m}\}$ is discarded from memory for the first time. Let $A_1 = \{\omega \in \Omega : T'_\omega \le L\}$ be the set of reward realizations for which $T'_\omega \le L$. Now fix some arm in $\{\texttt{arm}_{km+1}, \texttt{arm}_{km+2}, \ldots, \texttt{arm}_{km+m}\}$ say $\texttt{arm}_{km+i}$. Define $A_2^i = \{\omega \in \Omega : \texttt{arm}_{km+i} \in S_{T'_\omega}\}$ to be the event that the $\texttt{arm}_{km+i}$ belongs to $S_{T'_\omega}$. Now, let $A^i = A_1 \cap A_2^i$ be the set of reward realizations such that $\forall \omega \in A^i$, $T'_\omega \le L$ and $\texttt{arm}_i$ is discarded from memory before $Z_k$ becomes $T'_\omega + 1$. Also, for any event $A \subseteq \Omega$, let $\overline{A} = \Omega \setminus A$.

Now we have the following observation for permutation $\mathcal{I}'_2$.

**Observation 5.** *If $\omega \in \overline{A_1}$ and $\mathcal{P} \in \mathcal{I}'_2$ is an input permutation to the algorithm $\mathcal{A}$, then* $\sum_{i=1}^m X_{km+i} \ge \Omega(\frac{1}{m^2\varepsilon^2})$.

Let $i' = \arg\max_{i \in [m]} P_{0,R'}(A^i)$. We obtain the next observation due to the fact that the best arm $\texttt{arm}_{km+i'}$ is sampled for at most $L = o(T)$ times.

**Observation 6.** *If $\omega \in A^{i'}$ and $\mathcal{I}_1^{km+i'}$ is an input permutation to the algorithm $\mathcal{A}$, then $X_{\mathcal{I}_1^{km+i'}} \ge \frac{\varepsilon(T-T'_\omega)}{2} = \Omega(\varepsilon T)$.*

Now we will prove the following inequality which will be useful in our analysis:

$$m \cdot P_{i',R'}(A^{i'}) + P_{0,R'}(\overline{A_1}) \ge \frac{1}{4}. \tag{3}$$

The above inequality is trivially true if $P_{0,R'}(\overline{A_1}) \ge 1/4$. Therefore, let us assume $P_{0,R'}(\overline{A_1}) \le 1/4$, i.e., $P_{0,R'}(A_1) \ge 3/4$. Then $P_{0,R'}(A^{i'}) \ge 3/(4m)$, by averaging argument. Using Theorem 1 for

distributions $P_{0,R'}$ and $P_{i',R'}$, we obtain:

$$2(P_{0,R'}(A^{i'}) - P_{i',R'}(A^{i'}))^2$$
$$\leq \mathrm{KL}(P_{0,R'}, P_{i',R'}) \qquad\qquad\qquad \text{(by Pinsker's inequality)}$$
$$= \sum_{i\in[m]}\sum_{t=1}^{L} \mathrm{KL}(P_{0,R'}^{i,t}, P_{i',R'}^{i,t}) \qquad\qquad \text{(by chain rule)}$$
$$= \sum_{i\in[m]\setminus\{i'\}}\sum_{t=1}^{L} \mathrm{KL}(P_{0,R'}^{i,t}, P_{i',R'}^{i,t}) + \sum_{t=1}^{L} \mathrm{KL}(P_{0,R'}^{i',t}, P_{i',R'}^{i',t})$$
$$\leq 0 + L\cdot 2\varepsilon^2.$$

In the last inequality, the first term of the summation is zero because all arms $\mathtt{arm}_{km+i}$, where $i\in[m]\setminus\{i'\}$, have identical reward distributions under instances $\mathcal{P}\in\mathcal{I}_2'$ and $\mathcal{I}_1^{km+i'}$. To bound the second term in the summation, we use the last property from Theorem 1. Thus we have, $P_{0,R'}(A^{i'}) - P_{i',R'}(A^{i'}) \leq \varepsilon\sqrt{L}$. Hence, $P_{i',R'}(A^{i'}) \geq P_{0,R'}(A^{i'}) - \varepsilon\sqrt{L} \geq (3/(4m)) - (1/(2m)) = 1/(4m)$. Here, we use $P_{0,R'}(A^{i'}) \geq 3/(4m)$ and $L = 1/(4m^2\varepsilon^2)$. Hence, if $P_{0,R'}(\overline{A_1}) \leq \frac{1}{4}$ then $m\cdot P_{i',R'}(A^{i'}) \geq \frac{1}{4}$. This proves Inequality (2). Let $\mathcal{P}$ be an input permutation. Now we have the following:

$$\mathbb{E}[Y_k|R']$$
$$\geq \mathbb{P}[\mathcal{P}=\mathcal{I}_1^{km+i'}|R']\cdot P_{i',R'}(A^{i'})\cdot\Omega(\varepsilon T)$$
$$\quad + \mathbb{P}[\mathcal{P}\in\mathcal{I}_2'|R']\cdot P_{0,R'}(\overline{A_1})\cdot\Omega\Big(\frac{1}{m^2\varepsilon^2}\Big)$$
$$= \frac{\mathbb{P}[R'|\mathcal{P}=\mathcal{I}_1^{km+i'}]\cdot\mathbb{P}[\mathcal{P}=\mathcal{I}_1^{km+i'}]}{\mathbb{P}[R']}\cdot P_{i',R'}(A^{i'})\cdot\Omega(\varepsilon T)$$
$$\quad + \frac{\mathbb{P}[R'|\mathcal{P}\in\mathcal{I}_2']\cdot\mathbb{P}[\mathcal{P}\in\mathcal{I}_2']}{\mathbb{P}[R']}\cdot P_{0,R'}(\overline{A_1})\cdot\Omega\Big(\frac{1}{m^2\varepsilon^2}\Big)$$
$$\text{(Due to Bayes' Theorem )}$$
$$= \frac{\mathbb{P}[R'|\mathcal{P}=\mathcal{I}_1^{km+i'}]}{\mathbb{P}[R']}\cdot\frac{1}{2n}\cdot m P_{i',R'}(A^{i'})\cdot\Omega(\varepsilon T/m)$$
$$\quad + \frac{\mathbb{P}[R'|\mathcal{P}\in\mathcal{I}_2']}{\mathbb{P}[R']}\cdot P_{0,R'}(\overline{A_1})\cdot\frac{1}{4}\cdot\Omega\Big(\frac{1}{m^2\varepsilon^2}\Big)$$
$$\geq \frac{\mathbb{P}[R'|\mathcal{P}=\mathcal{I}_1^{km+i'}]}{\mathbb{P}[R']}\cdot\Omega\Big(\frac{T^{2/3}}{m^{4/3}n^{2/3}}\Big)$$
$$\quad \cdot (m\cdot P_{i',R'}(A^{i'}) + P_{0,R'}(\overline{A_1}))$$
$$\text{(Due to the fact that } \mathbb{P}[R'|\mathcal{P}=\mathcal{I}_1^{km+i'}] = \mathbb{P}[R'|\mathcal{P}\in\mathcal{I}_2']$$
$$\text{and } \Omega\Big(\frac{1}{m^2\varepsilon^2}\Big) = \Omega\Big(\frac{\varepsilon T}{mn}\Big) = \Omega\Big(\frac{T^{2/3}}{m^{4/3}n^{2/3}}\Big)$$
$$\geq \frac{\mathbb{P}[R'|\mathcal{P}=\mathcal{I}_1^{km+i'}]}{\mathbb{P}[R']}\cdot\Omega\Big(\frac{T^{2/3}}{m^{4/3}n^{2/3}}\Big)$$
$$\text{(Due to Inequality (3)).}$$

Therefore we now have the following:

$$
\mathbb{E}[Y_k] \geq \sum_{R' \in \mathcal{R}} \mathbb{P}[R'] \cdot \mathbb{E}[Y_k | R']
$$

$$
\geq \sum_{R' \in \mathcal{R}} \mathbb{P}[R' | \mathcal{P} = \mathcal{I}_1^{km+i'}] \cdot \Omega\Big(\frac{T^{2/3}}{m^{4/3}n^{2/3}}\Big)
$$

$$
\geq \Omega\Big(\frac{T^{2/3}}{m^{4/3}n^{2/3}}\Big).
$$

Let $\ell = \lfloor \frac{n}{2m} \rfloor - 1$. Then we have the following. :

$$
\mathbb{E}[R(t)] \geq \sum_{k=0}^{\ell} \mathbb{E}[Y_k]
$$

$$
\geq \sum_{i=0}^{\ell} \Omega\Big(\frac{T^{2/3}}{m^{4/3}n^{2/3}}\Big)
$$

$$
\geq \Omega\Big(\frac{n^{1/3}T^{2/3}}{m^{7/3}}\Big).
$$

$\square$

## C  Important Inequalities

**Lemma 12.** *(Hoeffding's inequality). Let $Z_1, \ldots, Z_n$ be independent bounded variables with $Z_i \in [0,1]$ for all $i \in [n]$. Then*

$$
\mathbb{P}\Big(\frac{1}{n}\sum_{i=1}^{n}(Z_i - \mathbb{E}[Z_i]) \geq t\Big) \leq e^{-2nt^2}\Big), \text{ and}
$$

$$
\mathbb{P}\Big(\frac{1}{n}\sum_{i=1}^{n}(Z_i - \mathbb{E}[Z_i]) \leq -t\Big) \leq e^{-2nt^2}\Big), \text{ for all } t \geq 0.
$$

**Theorem 13** (Berry-Esseen Theorem)**.** *There exists a positive constant $C \leq 1$ such that if $X_1, X_2, \ldots, X_n$ are i.i.d. random variables with $\mathbb{E}[X_i] = 0$, $\mathbb{E}[X_i^2] = \sigma^2 > 0$, and $\mathbb{E}[|X_i|^3] = \rho < \infty$, and if we define*

$$
Y_n = \frac{X_1 + X_2 + \ldots + X_n}{n}
$$

*to be the sample mean, with $F_n$ being the cumulative distribution function of $\frac{Y_n\sqrt{n}}{\sigma}$, and $\Phi$ be the cumulative distribution function of the standard normal distribution $\mathcal{N}(0,1)$, then for all $x$ and $n$,*

$$
|F_n(x) - \Phi(x)| \leq \frac{C\rho}{\sigma^3\sqrt{n}}.
$$

## D  Omitted Proofs from Section 4

**Lemma 7.** *The worst case sample complexity of the algorithm is $O(\frac{n}{\varepsilon^2} \cdot (\texttt{ilog}^{(r)}(n) + \ln(\frac{1}{\delta})))$ and it does not depend on the arms' rewards or gap parameter or the order in which the arms arrive.*

*Proof.* If $r = 1$, then the total number of samples is $n \cdot s_1 = O(\frac{n}{\varepsilon^2} \cdot (\texttt{ilog}^{(r)}(n) + \log(\frac{1}{\delta})))$. Let $c_0 := 1$. So for the rest of the analysis we assume that $r \geq 2$ and define $c_i := 2^{c_{i-1}}, \forall i > r$. Note that since $2 \leq r \leq \log^*(n)$, $c_2 \geq 2$ and $c_i = 2^{c_{i-1}}, \forall i \geq 3$. For any level $\ell - 1$, we send one arm from level $\ell - 1$ to level $\ell$ for every $c_{\ell-1}$ arms seen (this is excluding the arms sampled in Step 19). Hence, during the *Modified Selective Promotion*, the number of arms that can reach any level $\ell$ is at most $n/(\prod_{i=0}^{\ell-1} c_i)$. Each arm arriving at level $\ell$ is pulled exactly $s_\ell$ times. Also note that we can

sample up to $(r-1) \cdot s_r$ times in the Step 19. Since, we have $r$ levels, the total number of samples can be bounded as:

$$\sum_{\ell=1}^{r} \frac{n}{\prod_{i=0}^{\ell-1} c_i} \cdot s_\ell + (r-1) \cdot s_r$$

$$\leq \sum_{\ell=1}^{r} \frac{2n\beta_\ell\big(\mathtt{ilog}^{(r+1-\ell)}(n) + \log(\frac{2^{\ell+2}}{\delta})\big)}{\prod_{i=0}^{\ell-1} c_i} + r \cdot s_r$$

$$\leq n \cdot s_1 + r \cdot s_r$$

$$\quad + \frac{2n}{\varepsilon^2} \cdot \sum_{\ell=2}^{r} 4^{\ell+1} \cdot \left( \frac{\mathtt{ilog}^{(r+1-\ell)}(n)}{c_{\ell-1} \cdot c_{\ell-2}} + \frac{2\ell \cdot \log(\frac{2}{\delta})}{c_{\ell-1} \cdot c_{\ell-2}} \right)$$

$$\Big(\text{Since, } \prod_{i=0}^{\ell-1} c_i \geq c_{\ell-1} \cdot c_{\ell-2}, \beta_\ell = 4^{\ell+1}/\varepsilon^2, \text{ and}$$

$$\log(2^{\ell+2}/\delta) \leq \log(2^{2\ell}/\delta^{2\ell}) \leq 2\ell \log(2/\delta)\Big)$$

$$\leq n \cdot s_1 + r \cdot s_r$$

$$\quad + (2 \cdot 4^5 \cdot n/\varepsilon^2) \sum_{\ell=2}^{\infty} (4^{\ell-4}/c_{\ell-2})$$

$$\quad + (2^2 \cdot 4^5 \cdot n/\varepsilon^2) \cdot \log(2/\delta) \sum_{\ell=2}^{\infty} (4^{\ell-3}/c_{\ell-1})$$

$$\Big(\text{Since, } c_{\ell-1} = \mathtt{ilog}^{(r+1-\ell)}(n), \text{ and } (\ell/c_{\ell-2}) \leq 4\Big)$$

$$\leq n \cdot s_1 + r \cdot s_r + O(n/\varepsilon^2)\big(1 + \log(2/\delta)\big)$$

$$\Big(\text{Since, } \sum_{\ell=0}^{5} \frac{4^{\ell-2}}{c_\ell} = O(1), \sum_{\ell=6}^{\infty} \frac{4^{\ell-2}}{c_\ell} < \sum_{\ell=6}^{\infty} \frac{4^{\ell-2}}{8^{\ell-2}} < 1\Big)$$

$$\leq O(n/\varepsilon^2) \cdot (\mathtt{ilog}^{(r)}(n) + \log(1/\delta))$$

$$\Big(\text{Since, } r \cdot s_r = O(n/\varepsilon^2)(1 + \log(1/\delta)) \text{ and}$$

$$n \cdot s_1 = O(n/\varepsilon^2) \cdot (\mathtt{ilog}^{(r)}(n) + \log(1/\delta))\Big).$$

Hence, we have that the sample complexity is $O(\frac{n}{\varepsilon^2} \cdot (\mathtt{ilog}^{(r)}(n) + \log(\frac{1}{\delta})))$. $\qquad\square$

**Lemma 8.** *Let $\mathtt{arm}_1$ and $\mathtt{arm}_2$ be two different arms with means $\mu_1$ and $\mu_2$. Suppose $\mu_1 - \mu_2 \geq \theta$ and we sample each arm $\frac{K}{\theta^2}$ times to obtain empirical means $\widehat{\mu}_1$ and $\widehat{\mu}_2$. Then,*

$$\mathbb{P}(\widehat{\mu}_1 \leq \widehat{\mu}_2) \leq 2 \cdot e^{(-K/2)}$$

*Proof.*

$$\mathbb{P}(\widehat{\mu}_1 > \widehat{\mu}_2) \geq \mathbb{P}\Big(\mu_1 - \frac{\theta}{2} < \widehat{\mu}_1 \text{ and } \widehat{\mu}_2 < \mu_2 + \frac{\theta}{2}\Big)$$

$$= \mathbb{P}\Big(\mu_1 - \frac{\theta}{2} < \widehat{\mu}_1\Big) \cdot \mathbb{P}\Big(\widehat{\mu}_2 < \mu_2 + \frac{\theta}{2}\Big)$$

$$\geq \big(1 - e^{-2 \cdot \frac{K}{\theta^2} \cdot (\frac{\theta}{2})^2}\big) \cdot \big(1 - e^{-2 \cdot \frac{K}{\theta^2} \cdot (\frac{\theta}{2})^2}\big) \qquad \text{(Due to Hoeffding's Inequality)}$$

$$\geq (1 - 2 \cdot e^{-K/2}).$$

Hence, $\mathbb{P}(\widehat{\mu}_1 \leq \widehat{\mu}_2) \leq 2 \cdot e^{-K/2}$. $\qquad\square$

# E Adversarial Example for [4]

In this section, we show an adversarial example to show that the algorithm of [4] is not $(\varepsilon, \delta)$-PAC. First, we state the algorithm of [4] (Algorithm 2) here for completeness. The algorithm takes as input

$n \in \mathbb{N}$ arms arriving in a stream in an arbitrary order, an approximation parameter $\varepsilon \in [0, 1/2)$, and the confidence parameter $\delta \in (0, 1)$. We first define some notation that is used throughout this work, which is consistent with the notation used in [4].

$$\{r_\ell\}_{\ell=1}^\infty : \quad r_1 = 4; \; r_{\ell+1} = 2^{r_\ell}; \qquad \text{(intermediate parameters used to define } s_\ell \text{ below)}$$

$$\varepsilon_\ell = \varepsilon/(10 \cdot 2^{\ell-1}); \qquad \text{(intermediate estimate of gap parameter)}$$

$$\beta_\ell = 1/\varepsilon_\ell^2;$$

$$\{s_\ell\}_{\ell=1}^\infty : \quad s_\ell = 4\beta_\ell\big(\ln(1/\delta) + 3r_\ell\big); \qquad \text{(number of samples per arm in level } \ell)$$

$$\{c_\ell\}_{\ell=1}^{\lceil \log^* n \rceil + 1} : \quad c_1 = 2^{r_1}; \; c_\ell = 2^{r_\ell}/2^{\ell-1};$$

(the number of arms processed in level $\ell$ before

sending $\texttt{arm}_\ell^*$ to level $\ell + 1$)

For Algorithm 2 to be $(\varepsilon, \delta)$-PAC, it has to find an $\varepsilon$-best arm with probability at least $1 - \delta$. We next provide a formal argument for why Algorithm 2 is not an $(\varepsilon, \delta)$-PAC algorithm by providing a counterexample.

---

**Algorithm 2**

---

1: $\{r_\ell\}_{\ell=1}^\infty : r_1 := 4, r_{\ell+1} = 2^{r_\ell};$
2: $\varepsilon_\ell = \frac{\varepsilon}{10 \cdot 2^{\ell-1}};$
3: $\beta_\ell = \frac{1}{\varepsilon_\ell^2};$
4: $s_\ell = 4\beta_\ell\big(\ln(\frac{1}{\delta}) + 3r_\ell\big);$
5: $c_1 = 2^{r_1}, c_\ell = \frac{2^{r_\ell}}{2^{\ell-1}} (\ell \geq 2);$
6: Counters: $C_1, C_2, \ldots, C_t$ initialized to 0 where $t = \lceil \log^*(n) \rceil + 1.$
7: Stored arms: $\texttt{arm}_1^*, \texttt{arm}_2^*, \ldots, \texttt{arm}_t^*$ the most biased arm of $\ell$-th level.
8: **while** A new arm $\texttt{arm}_i$ arrives in the stream **do**
9:     Read $\texttt{arm}_i$ to memory
10:     **Aggressive Selective Promotion:** Starting from level $\ell = 1$:
11:     Sample **both** $\texttt{arm}_i$ and $\texttt{arm}_\ell^*$ for $s_\ell$ times.
12:     Drop $\texttt{arm}_i$ if $\hat{p}_{\texttt{arm}_i} < \hat{p}_{\texttt{arm}_\ell^*}$, otherwise replace $\texttt{arm}_\ell^*$ with $\texttt{arm}_i$.
13:     Increase $C_\ell$ by 1.
14:     If $C_\ell = c_\ell$, make $C_\ell$ equal to 0, send $\texttt{arm}_\ell^*$ to the next level by calling Line 11 with $(\ell = \ell + 1)$.
15: **end while**
16: Return $\texttt{arm}_t^*$ as the selected most bias arm.

---

Let the arms arrive in the stream be $\texttt{arm}_1, \ldots, \texttt{arm}_n$, where $n > (c_1 \cdot c_2)$. Let $\texttt{arm}_i$ be the $i^{th}$ arm to arrive in the stream and has a mean $p_i$, where $i \in [n]$. For all $i > c_1 \cdot c_2$, let $p_i = 0$. For all $i \leq c_1 \cdot c_2$, let $p_i = \frac{1}{2} - (\lceil \frac{i}{c_1} \rceil - 1) \cdot \frac{\varepsilon}{c_2 - 2}$. Let $\texttt{arm}_{k_1}, \texttt{arm}_{k_2}, \ldots, \texttt{arm}_{k_{c_2}}$ be the first $c_2$ arms which arrive at level 2 (note that all the arms which arrive at level 2 after $\texttt{arm}_{k_{c_2}}$ will have a mean of 0). Let $\texttt{arm}_2^*$ be the most biased arm (based on the sampling) at the end of Aggressive Selection Promotion step for level $\ell = 2$ for $\texttt{arm}_{k_{c_2}}$. Now $C_2 = c_2$ after the arrival of $\texttt{arm}_{k_{c_2}}$. Thus $\texttt{arm}_2^*$ will be sent to level 3. As all the following remaining arms have lesser means, at the end, the algorithm finally returns an arm with mean less than or equal to the mean of $\texttt{arm}_2^*$. Note that $\forall i \in [c_2]$, all the arms in the set $\{\texttt{arm}_{(i-1)\cdot c_1 + 1}, \ldots, \texttt{arm}_{i \cdot c_1}\}$ have the same mean and one among them is sent as $\texttt{arm}_{k_i}$ to level 2. Therefore $p_{k_i} = p_{k_1} - (i-1)\frac{\varepsilon}{b}$, where $b = 2^{15} - 2$ and $p_{k_1} = \frac{1}{2}$. So for any $i \in [c_2 - 1]$, $p_{k_i} - p_{k_{i+1}} = \frac{\varepsilon}{b}$. We will show that with probability $> \delta$, we send $\texttt{arm}_{k_{c_2}}$ to level 3, and $\frac{1}{2} - p_{k_{c_2}} > \varepsilon$.

For $i \in [c_2]$, let $Y_i^t$ denote the reward when we sample the arm $\texttt{arm}_{k_i}$ for the $t$-th time. We assume that $Y_i^t \sim \text{Bern}(p_{k_i})$ and $\text{Var}[Y_i^t] = p_{k_i}(1 - p_{k_i})$ (Note that this is a reasonable assumption as the Algorithm 2 should work for any distribution). For $i \in [c_2 - 1]$, let $Z_i^t = Y_i^t - Y_{i+1}^t$. Clearly, $\mu_i := \mathbb{E}[Z_i^t] = \frac{\varepsilon}{b}$. Let $\sigma_i^2 := \text{Var}[Z_i^t]$. Let us assume that $\varepsilon < \frac{1}{5}$ (Later we will choose $\varepsilon$ in such a way so that this condition is satisfied). In this case, $\sigma_i^2 = \text{Var}[Y_i^t] + \text{Var}[Y_{i+1}^t] > 2(p_{k_{c_2}})(1 - p_{k_{c_2}}) > \frac{2}{5}$. Let $Z_i = Z_i^1 + Z_i^2 + \ldots + Z_i^{s_2}$. Note that, if every arm from the set $\texttt{arm}_{k_1}, \texttt{arm}_{k_2}, \ldots, \texttt{arm}_{k_{c_2}}$

when it arrives in the level 2 beats $\mathtt{arm}_2^*$ in the challenge, then $\mathtt{arm}_{k_{c_2}}$ will be sent to level 3. Thus, $\{Z_i < 0, \forall i \in [c_2 - 1]\} \subseteq \{\mathtt{arm}_{k_{c_2}} \text{ is sent to level 3}\}$.

Assuming that $\delta$ and $\varepsilon$ are very small (which we will choose appropriately to bound the error), we approximate (using the central limit theorem) the distribution of $Z_i$ using the normal distribution $\mathcal{N}(s_2\mu_i, s_2\sigma_i^2)$. Hence,

$$\mathbb{P}[Z_i < 0] = \mathbb{P}[Z_i > 2s_2\mu_i]$$
$$= 1 - \frac{1}{2}\left(1 + \mathrm{erf}\left(\frac{s_2\mu_i}{\sqrt{2s_2\sigma_i^2}}\right)\right)$$

Hence, we have

$$\mathbb{P}[Z_i < 0] = \frac{\mathrm{erf}\left(\frac{s_2\mu_i}{\sqrt{2s_2\sigma_i^2}}\right)}{2}$$
$$\geq \frac{\mathrm{erf}\left(s_2\mu_i \cdot \sqrt{\frac{5}{4s_2}}\right)}{2} \qquad \text{(Since, } \sigma_i^2 \geq \frac{2}{5} \text{ and } \mathrm{erf}(x) \text{ is decreasing in } x)$$
$$= \frac{\mathrm{erf}\left(\frac{20\sqrt{5}}{b}\sqrt{\ln\left(\frac{e^{3r_2}}{\delta}\right)}\right)}{2} \qquad \text{(Substituting } s_2 = 4\beta_2\left(\ln(\frac{1}{\delta}) + 3r_2\right))$$
$$\geq \frac{\sqrt{\gamma - 1}}{2}e^{-\frac{2000 \cdot \gamma}{b^2}\ln\left(\frac{e^{3r_2}}{\delta}\right)}$$
$$\qquad \text{(Since, } \mathrm{erf}(x) \geq (\gamma - 1)^{1/2}e^{-\gamma x^2}, \forall x \geq 0, \ \gamma := \sqrt{2e/\pi})$$
$$= \frac{\sqrt{\gamma - 1}}{2}\left(\frac{\delta}{e^{3r_2}}\right)^{\frac{2000 \cdot \gamma}{b^2}}.$$

Thus, we can lower bound the probability that $\mathtt{arm}_{k_{c_2}}$ is sent to level 3 as follows:

$$\mathbb{P}[\mathtt{arm}_{k_{c_2}} \text{ is sent to level 3}]$$
$$\geq \mathbb{P}[Z_i < 0, \forall i \in [c_2 - 1]]$$
$$= \prod_{i \in [c_2 - 1]} \mathbb{P}[Z_i < 0] \qquad \text{(Since, the arm pulls are independent)}$$
$$\geq \left(\frac{\sqrt{\gamma - 1}}{2}\left(\frac{\delta}{e^{3r_2}}\right)^{\frac{2000 \cdot \gamma}{b^2}}\right)^{c_2 - 1}$$
$$= \frac{\delta^{\frac{2000 \cdot \gamma \cdot (c_2 - 1)}{b^2}}}{K}, \text{ where } K = \left(\frac{2 \cdot e^{\frac{3r_2 \cdot 2000 \cdot \gamma}{b^2}}}{\sqrt{\gamma - 1}}\right)^{c_2 - 1}.$$

Consider that function $f(x) = \frac{e^{x\left(1 - \frac{2000 \cdot \gamma \cdot (c_2 - 1)}{b^2}\right)}}{K}$. Since, $f(x)$ is an increasing and convex function, there is a constant $c$ such that $f(c) > 2$. This implies that for $\delta = e^{-c}$ we have the following:

$$\mathbb{P}[\mathtt{arm}_{k_{c_2}} \text{ is sent to level 3}] \geq \frac{\delta^{\frac{2000 \cdot \gamma \cdot (c_2 - 1)}{b^2}}}{K}$$
$$= f(c)e^{-c}$$
$$> 2\delta.$$

Now, we bound the error in calculation of the above probability. Using the Berry-Esseen theorem, the error $\varepsilon_i$ of calculating $\mathbb{P}[Z_i < 0]$ is upper bounded by $\frac{C\rho}{\sigma_i^3\sqrt{s_2}} \leq \frac{\varepsilon}{\sqrt{\ln(\frac{1}{\delta})}}$, where $C \leq 1$, $\rho = \mathbb{E}[|Z_i^t - \mu_i|^3] \leq 8$ (as $|Z_i^t - \mu_i| \leq 2$) and $\sigma_i^2 = \mathrm{Var}[Z_i^t - \mu_i] = \mathrm{Var}[Z_i^t]$. Also we assumed $\mathrm{Var}[Z_i^t] > \frac{2}{5}$ (we will choose $\varepsilon$ in such a way that this is satisfied). If we choose $\varepsilon$ such that

$\varepsilon < \frac{\delta\sqrt{\ln(\frac{1}{\delta})}}{c_2}$ and $\varepsilon < 1/5$, then $\varepsilon_i < \frac{\delta}{c_2}$. Hence, we can conclude that $\mathbb{P}[\mathtt{arm}_{k_{c_2}}$ is sent to level 3$] >$ $2\delta - \sum_{i=1}^{c_2-1} \varepsilon_i > 2\delta - \delta = \delta$.

As $\frac{1}{2} - p_{k_{c_2}} = \left(\frac{c_2-1}{c_2-2}\right) \cdot \varepsilon$, we can conclude that with probability $> \delta$, the Algorithm 2 returns an arm with reward gap $> \varepsilon$ from the best arm.