# OpenReview forum: "Multi-Armed Bandits with Bounded Arm-Memory: Near-Optimal Guarantees for Best-Arm Identification and Regret Minimization"
_NeurIPS.cc/2021/Conference — NeurIPS 2021 Poster_

### Official Review · Reviewer_QqCn · 2021-07-07

**Rating:** 8
**Confidence:** 3

**Summary:**

This paper studies the classical $T$-period $n$-armed stochastic bandit problem under "memory" constraints: the decision maker can only store reward statistics for a certain maximum number $m < n \ll T$ of arms at any time $t\in[T]$. The authors establish an $\Omega\left( T^{2/3} \right)$ fundamental hardness result (modulo dependence on $m,n$) for the cumulative regret minimization problem under "single-pass" policies (arms discarded from memory cannot be recalled) with memory sizes up to $n-1$. This is optimal in $T$ since an algorithm with a matching upper bound in $T$ (up to log factors) already exists in literature. The result is also particularly interesting since it is known that the "holy grail" of $\Theta\left( \sqrt{T} \right)$ regret (modulo $n$) is achievable with memory size $n$. The authors also study the memory-limited best arm identification problem under the fixed confidence setting, and propose an $\left( \epsilon,\delta\right)$-PAC algorithm (where the parameters are the approximation and confidence factors respectively) with a space complexity of $m = \mathcal{O}\left( \log^\ast n \right)$ and an optimal sample complexity. Notably, a different algorithm that was claimed to have these very properties, already appeared in a STOC'20 paper. The authors show that the claims made in said paper are, in fact, erroneous, by constructing a family of instances on which aforementioned properties are provably violated.

**Limitations And Societal Impact:**

Though I did not verify each and every step of the proofs very rigorously, the authors provide plenty of intuition and the proofs generally appear correct. The literature review seems to be comprehensive too, though there are a couple of papers I would like to suggest. I recently discovered these works on a related infinite-armed formulation of the stochastic bandits problem (https://arxiv.org/pdf/2105.10721.pdf and https://arxiv.org/pdf/2103.12452.pdf). Cited works show that it is possible to achieve a logarithmic instance-dependent regret if there exists a constant fraction of arms of the optimal "type," and the best and second-best arm-types are "well-separated" in the mean-reward, despite an infinite population of arms. More interestingly and relevant to this submission, it is shown that the logarithmic bound can be achieved by algorithms that only store a finite number of arms at any time (as in https://arxiv.org/pdf/2105.10721.pdf), or at worst, logarithmic in the horizon (https://arxiv.org/pdf/2103.12452.pdf). I think these could enrich the discussion.

**Main Review:**

The paper is well-written overall with accessible proofs. I really enjoyed reading it. The $\Omega\left( T^{2/3} \right)$ lower bound for single-pass algorithms applies also to a random-order arrival of arms (not just adversarial), which is interesting. I think the paper will certainly add value to the literature in this area.

**Time Spent Reviewing:**

8

---

> ### Author Response · Authors · 2021-08-09
> **Response to Reviewer QqCn**
>
> We thank the reviewer for pointing us to the additional literature which will enrich our discussion. It was very helpful and we will discuss these in the updated version of our paper.

---

### Official Review · Reviewer_h7sF · 2021-07-09

**Rating:** 7
**Confidence:** 4

**Summary:**

This paper studies a variant of stochastic multi-armed bandits in which the learner doesn't have access to all the arms. The learner has a memory of size $m$, which means it can store and access only $m$ arms. The learner can discard some arms to free up space for new arms, which come in a stream. When an arm is discarded, it can never be accessed again.

The standard problems of regret minimization and best-arm identification are studied here; a regret lower bound for the former and an algorithm for the latter are given.

Let $T, n,$ and $m$ denote the time horizon, the number of arms, and the memory size, respectively. When $m=2$, a regret upper bound of $\widetilde{O}(n^{1/3}T^{2/3})$ was shown in previous work. This paper gives a matching lower bound up to logarithmic factors, and gives a lower bound of ${\Omega}(n^{1/3}T^{2/3}/m^{7/3})$ for all $m=1,2,\dots,n-1$. This is in sharp contrast with the $m=n$ case, which achieves $O(\sqrt{nT})$ regret.

For best-arm identification, the paper provides an algorithm with memory size $1 + \log^* (n)$ that pulls $O(n\log(1/\delta)/\varepsilon^2)$ arms and outputs an $\varepsilon$-optimal arm with probability $1-\delta$. This sample complexity is optimal even with memory size $n$. A similar result was shown in a previous work, but this paper claims that that proof had a bug.

**Ethical Concerns:**

None.

**Limitations And Societal Impact:**

Not applicable.

**Main Review:**

The problem is natural and well-motivated, and the results are strong, so I recommend acceptance.

One surprising implication of this paper is that having limited memory significantly affects regret-minimization but doesn't affect best-arm identification very much.

I have checked some of the proofs (those that appear in the body) and they seem correct. The inclusion of Theorem 2, which is a weaker version of the paper's lower bound was a good idea. One part of one proof should be adjusted, as I will explain below.

The paper claims to fix a bug from a STOC'20 paper. While I haven't read that paper and can't verify this claim, I think there is great value in fixing bugs of previous work.

The paper can be polished a lot more, and I will add some comments below. But it's acceptable as a conference paper.

Detailed comments and suggestions:
- Don't confuse arm-memory size and space complexity. It makes the paper confusing. Space complexity of an algorithm is a well-defined concept. Your result concern arm-memory size, which has connections with space complexity, but they're not the same. For example, in line 164 you say that UCB1 has space complexity $\Omega(n)$. But that is irrelevant. What matters is that the arm-memory size is $n$ for UCB1. The constants are important here. Your result holds for arm-memory size smaller than $n$ regardless of the space complexity. That is, even if there is no limit on space complexity but the arm-memory size is smaller than $n$, your lower bound holds. I suggest that you stick to expressing your results in terms of arm-memory size, and don't use $O, \Omega$ notation because the constants matter for your lower bound. You can have a remark explaining the connection with space complexity.

- The union bound used in Claim 1 and Claim 2 should be changed. Here, you say you are using Lemma 8, which concerns pairs of arms. If you're gonna do a union bound over the pairs, that would give you $c_{\ell}^2$ events to apply the union bound to. But this would shatter your calculations. What you want to do is to replace Lemma 8 with a lemma stating that each arm has been estimated within certain distance with certain probability, and then you can apply the union bound over the arms rather than pairs of arms.

- The paper title is misleading as you don't give guarantees for regret minimization. A lower bound is not a guarantee.

- Line 72: Your lower bound matches the upper bound *only when $m=2$.* This should be clarified.

- Line 76: The actual dichotomy is between $m=n-1$ and $m=n$. This should be clarified.

- Define the $\widetilde{O}$ notation.

- Line 110: "They do not provide a regret bound for their algorithm, hence their regret can become very high." I don't understand the deduction here. If I can't prove that regret is small doesn't mean that regret is large.

- Line 136: You say an MAB instance is specified by the arm means. But the distributions are also important, no?

- Line 140: It seems you're restricting the algorithm to store only the mean and the number of pulls for each arm. But your proof doesn't require that; it seems your lower bound is general, regardless of what the algorithm stores. So why do you put such a restriction in your model?

- Line 161: "distribution" should be "randomization" or "realization"

- Line 187: "We settle the question by showing ..." You don't really settle the question. You prove a lower bound. Settling the question would mean proving matching lower and upper bounds for all $m$.

- Line 212 to 219: This is not clearly explained. You can just say "The algorithm doesn't lose anything by first loading the first $m$ arms into its memory." This is intuitive enough that all readers will believe you.

- Line 221: Tuple is not a correct word here; use table or sequence.

- Line 297: "the algorithm returns the arm in the highest level." This is not true. The post-processing step is skipped here, which confuses the reader.

- Algorithm 1: The values of $\varepsilon_{\ell}$ and $\beta_{\ell}$ are never used in the algorithm, so why define them in the pseudocode? Also, the comment of line 4 is very confusing; because the number of arms in level $\ell$ is either 0 or 1.

- Statement of Claim 1 is confusing. When you say "an arm is sent," do you mean some arm, all arms, or there exists at least one arm? Similarly for Claim 2, and the proof of Lemma 9. The phrase "an arm" can mean at least one arm or all arms.

- Statement of Claim 1, "an arm with reward gap at most $\varepsilon$ from arm'" is understandable but can be made easier to understand.

**Time Spent Reviewing:**

5

---

> ### Author Response · Authors · 2021-08-09
> **Response to Reviewer h7sF**
>
> We thank the reviewer for the detailed feedback and excellent insightful suggestions which will be very helpful to us in improving the presentation of our work. We will incorporate these suggestions in the updated version of our work. We address the reviewer queries below.
>
> Reviewer's concern: “The union bound used in Claim 1 and Claim 2 should be changed. Here, you say you are using Lemma 8, which concerns pairs of arms. If you're gonna do a union bound over the pairs, that would give you $c_\ell^2$  events to apply the union bound to. But this would shatter your calculations. What you want to do is to replace Lemma 8 with a lemma stating that each arm has been estimated within certain distance with certain probability, and then you can apply the union bound over the arms rather than pairs of arms.”
>
> Author's response: While using the union bound In the proof of Claim 1, we are comparing the mean of each of the $c_\ell$ arms with the mean of the best arm (see Line 355). So the number of events is $c_\ell$ and not $c_\ell^2$. Hence, our calculations are correct to use the union bound. Similar reasoning holds true for Claim 2 and hence the current statement of Lemma 8 suffices for our proofs.
>
> Reviewer's concern: “They do not provide a regret bound for their algorithm, hence their regret can become very high." I don't understand the deduction here. If I can't prove that regret is small doesn't mean that regret is large.”
>
> Author's response:  Thanks for the comment. What we meant by the statement is the following: The authors provide an instance-dependent bound with an inverse dependence on $\Delta_i$’s (the gap parameter). If the $\Delta_i$ is of the order of say $(1/T^{2/3})$ then the proven regret bound implies $O(T^{2/3})$ regret. Since an instance-independent worst-case regret bound is not provided, there is no tighter regret guarantee than what can be inferred from the instance-dependent regret expression.
>
> Reviewer's concern: “Line 140: It seems you're restricting the algorithm to store only the mean and the number of pulls for each arm. But your proof doesn't require that; it seems your lower bound is general, regardless of what the algorithm stores. So why do you put such a restriction in your model?”
>
> Author's response: Thank you for the suggestion. Yes, our result indeed holds without this restriction.
>
> Reviewer's concern:  “Line 136: You say an MAB instance is specified by the arm means. But the distributions are also important, no?”
>
> Author's response:  We study the MAB problem where the distribution is characterized only by its mean.
>
>
> Reviewer's concern: “Algorithm 1: The values of  $\varepsilon_\ell$ and $\beta_\ell$ are never used in the algorithm, so why define them in the pseudocode? Also, the comment of line 4 is very confusing; because the number of arms in level  is either 0 or 1.”
>
> Author's response:  $\varepsilon_\ell$ is used in defining $\beta_\ell$ (Line 2 of Algo) and is also used in the proofs in Claim 1,2 and Lemma 9. $\beta_\ell$ has been used in defining $s_\ell$ (Line 3 of Algo) and proof of Lemma 7. In Comment of line 4, the reviewer is correct in saying that the number of arms stored at any level is 0 or 1, but by the number of arms in level $\ell$  we mean that $c_\ell$ is the number of arms that arrive in level $\ell$ before we send arm$^*_\ell$  to the level $\ell+1$.
>
> Reviewer's concern: “Statement of Claim 1 is confusing. When you say "an arm is sent," do you mean some arm, all arms, or there exists at least one arm? Similarly for Claim 2, and the proof of Lemma 9. The phrase "an arm" can mean at least one arm or all arms.”
>
> Author's response: In Claim 1, when we say “an arm is sent”, we mean that there exists at least one arm in level $\ell$ which is sent to the level $\ell+1$. The same thing holds true for Claim 2 and proof of Lemma 9.

---

> > ### Comment · Reviewer_h7sF · 2021-08-27
> > **Thanks**
> >
> > Thanks for your answers. Please clarify the concerns in the updated version.
> >
> > I still have one comment. I don't understand the sentence in the rebuttal that says "we study the MAB problem where the distribution is characterized only by its mean." An instance of an MAB problem is characterized by distributions, and a distribution IS NOT characterized by its mean. If you want the mean to characterize the distribution, you can, for example, consider special classes of distributions, e.g., Bernoulli distributions. Also, add this clarification to the paper, and discuss whether your results generalize to general distributions.

---

> > > ### Author Response · Authors · 2021-08-28
> > > **Thanks**
> > >
> > > We thank the reviewer for the comment. We will clarify and incorporate the suggested changes in the updated version. Indeed, MAB instance is specified by the distributions corresponding to the arms and not just the means of the distributions. In fact, the knowledge of the distributions is not required in our analysis, and in particular, our analysis holds for general distributions.

---

### Official Review · Reviewer_8qyp · 2021-07-14

**Rating:** 6
**Confidence:** 3

**Summary:**

The paper studied the multi-armed bandit problem with bounded arm-memory. In this model, at each time at most $m<n$ arms (and their statistics) may be stored in memory.
The paper is focused on a single-pass model, in which after an arm being removed from the memory it can no longer be accessed.

For the regret minimization problem, the authors prove a $\Omega(n^{1/3}T^{2/3}/m^{7/3})$ lower bound, and show a simple $\tilde{O}(n^{1/3}T^{2/3})$ upper bound.

For the best arm identification problem, they show an algorithm with sample complexity of $O(\frac{n}{\epsilon^2}(\log(1/\delta)+ilog^{(m-1)}(n)))$ where $ilog^{(r)}$ is the iterated logarithm of order $r$.
Thus showing that $log^*(n)$ memory is sufficient for asymptotically optimal sample complexity.
They also point out an error in [1] who previously claimed for showing such an algorithm.


**Limitations And Societal Impact:**

Any societal impact depends on the application.

**Main Review:**

The model is a very natural variant of the multi-armed bandit problem and is interesting both from the theoretical perspective and the possible practical uses.
The authors present two separate results that each is limited by its own and I’m not sure that tying them together is the right way to overcome this issue.

Questions and comments:

Is there an upper bound for the regret minimization problem that does use additional memory for $m>2$?
What is the known lower bound if we allow several passes?

The algorithm for the best arm identification problem seems close to the problem of MAB using a few rounds of sampling (where only a small number of arms are qualified for the next round). [2] studied the problem and showed lower bounds on the number of rounds needed for optimal sample complexity. On the other hand, under the assumption that $\delta<1/n$, [3] show that the optimal sample complexity can be reached using a constant number of rounds. Those techniques may be useful for your model as well.

[1] Sepehr Assadi and Chen Wang. Exploration with limited memory: streaming algorithms for
coin tossing, noisy comparisons, and multi-armed bandits. In Proceedings of the 52nd Annual
ACM SIGACT Symposium on Theory of Computing, pages 1237–1250, 2020.

[2] Arpit Agarwal, Shivani Agarwal, Sepehr Assadi, Sanjeev Khanna. Learning with Limited Rounds of Adaptivity: Coin Tossing, Multi-Armed Bandits, and Ranking from Pairwise Comparisons. In Proceedings of the 2017 Conference on Learning Theory, PMLR 65:39-75, 2017.

[3] Avinatan Hassidim, Ron Kupfer, Yaron Singer. An Optimal Elimination Algorithm for Learning a Best Arm. Advances in Neural Information Processing Systems 33 (2020): 10788-10798.‏


**Time Spent Reviewing:**

6

---

> ### Author Response · Authors · 2021-08-09
> **Response to Reviewer 8qyp**
>
> Reviewer's concern: “Is there an upper bound for the regret minimization problem that does use additional memory for m>2?”
>
> Author's response: We are not aware of any single-pass algorithm that uses m>2 arms. However, note that the algorithm in L178-185 achieves optimal (in n and T) single-pass regret while storing only two arms in memory (m = 2). As this is already optimal for the single-pass setting, we cannot hope to get a better regret guarantee in n and T with m > 2 arms in the memory.
>
>
> Reviewer's concern: “What is the known lower bound if we allow several passes?”
>
> Author's response: There is a $\Omega(\sqrt{T})$ lower bound for the offline setting.  $O(\sqrt{T})$ upper bound is achieved by a log T pass algorithm by RoyChaudhuri and Kalyanakrishnan [AAAI2020] (citation 24 in our submission). The problem of establishing a lower bound where the number of passes is less than $\log T$ is open and could be interesting future work.
>
> Reviewer's suggestion: “The algorithm for the best arm identification problem seems close to the problem of MAB using a few rounds of sampling (where only a small number of arms are qualified for the next round). [2] studied the problem and showed lower bounds on the number of rounds needed for optimal sample complexity. On the other hand, under the assumption that  δ<1/n, [3] show that the optimal sample complexity can be reached using a constant number of rounds.”
>
> Author's response: We thank the reviewer for pointing out this additional literature to enrich the discussion. We would also like to point out that our best arm identification algorithm is a streaming version of the r-round adaptive algorithm, where the arm pulls in each round are decided based on the observed outcomes in the earlier rounds. The best-arm is the output at the end of r rounds. The upper bound on the sample complexity of our algorithm matches with the lower bound for any r-round adaptive algorithm which are mentioned in the paper [Agarwal et al., COLT’17] referred by the reviewer.

---

### Official Review · Reviewer_EMgB · 2021-07-15

**Rating:** 5
**Confidence:** 3

**Summary:**

The authors consider the problem of multi-armed bandits with the additional constraint that the learner can only pull arms that belong to the so-called arm-memory. First, the authors look at the regret minimization setting, and provide a worst-case regret lower bound of order $\Omega(T^{3/2})$ that any single-pass algorithm must satisfy, given that arm-memory size is strictly smaller than the number of arms. Single-pass algorithms can only add an arm or remove it from the arm-memory once. Second, the authors investigate the pure exploration problem and propose an algorithm that is $(\epsilon,\delta)$-PAC, only uses $O(\log^\star(n))$ space complexity, and has a sample complexity of order $O(\frac{n}{\epsilon^2} \log(1/\delta))$.

**Ethical Concerns:**

I do not see any ethical issues with the current algorithm. The work is mainly theoretical.

**Limitations And Societal Impact:**

I do not see any potential negative impact of this work. The work is mainly theoretical.

**Main Review:**

The arrival of arms is poorly explained. Can the authors clarify how do arms arrive in detail? is there a delay between the arms' arrival? is there a probability distribution that governs the arms' arrival? Shouldn't that impact, for instance, the regret lower bound, if the best arm has a low probability of arrival? Furthermore, the authors present Theorem 1, 2, and 3 but all these theorems have the same statement! The authors should state their results in detail.

Existing regret upper bounds have a worst-case regret upper bound of order $\tilde{O}(\sqrt{T}$, while the provided lower bound is of order $\Omega(T^{3/2})$. It can be understood from the authors' result that this is because the class of algorithm over which the lower bound holds is restricted to a single-pass algorithm. Could the author confirm this and provide a discussion on this? Could they also explain how does the single-pass restriction appear in the proof? Furthermore, the algorithm would ensure regret only if the best arm is not played, discarding a suboptimal arm from the memory does not necessarily affect regret. Could the authors explain why observations 1 and 2 hold? for instance according to observation 1, if $\bar{A_1}$ holds, then $T'> L$, and simply means that the first time algorithm ${\cal A}$ discards an arm is larger than $L$. It could be a sub-optimal arm, in which case it wouldn't matter or maybe I am missing something. Furthermore, why was $\epsilon$ chosen to be of order $1/T^{1/3}$, and $L$ chosen to be of order $1/\epsilon^2$? These choices seem arbitrary, but they are behind the lower bound of order $T^{2/3}$.  Could the authors explain the intuition behind these choices?


For the pure exploration problem, the authors' algorithm is proven to be $(\epsilon,\delta)$-PAC and optimal in a mini-max sense (although they do not provide a lower bound) using only $O(\log^\star(n))$. It is not surprising that the memory requirement for best arm identification is going to be small, nonetheless, it is interesting to have an algorithm with such a guarantee. However, the minimax guarantee is rather disappointing and does not provide truly an insight into the trade-off of the sample complexity vs. memory complexity. Looking into algorithms that are instance-dependent optimal makes more sense. In fact, one can note that finding a minimax optimal algorithm that is $\delta$-PAC is an ill-posed problem since one can easily show that the lower bound in the minimax sense is infinite. Perhaps the authors should look at the paper entitled "Optimal Best Arm Identification with Fixed Confidence" by Garivier and Kaufmann.

**Time Spent Reviewing:**

17 hours

---

> ### Author Response · Authors · 2021-08-09
> **Response to the Reviewer EMgB**
>
> Reviewer's concern: “The authors present Theorem 1, 2, and 3 but all these theorems have the same statement! The authors should state their results in detail.”
>
> Author's response: We assume that the reviewer is referring to Theorems 2, 3, 4 and not Theorems 1, 2, 3 since Theorem 1 is a standard result related to KL divergence (see Chapter 2 from A. Slivkins). However, please note that even Theorems 2, 3, 4 do not have the same statements. Theorem 2 provides a weaker version of our lower bound (Theorem 3). As also noted by Reviewer h7sF, we have done this to provide a simpler exposition of the proof techniques used in our lower bound proof in the main body. Theorem 3 is our main lower bound result, the proof of which we provide in the appendix. Theorem 4 shows that this lower bound also holds for the random order arrival setting which is interesting as is also pointed out by Reviewer QqCn.
>
> Reviewer's concern: “These choices of L and $\varepsilon$ seem arbitrary… . Could the authors explain the intuition behind these choices?”
>
> Author's response: We would like to note that the choice of $\varepsilon$ and $L$ is not arbitrary. To prove a lower bound of X, we need to show that, given any algorithm, there is an instance on which it will suffer ‘X’ regret. Our instance has $\varepsilon$ as a parameter. Hence, we need to show that for some value of $\varepsilon$, the regret suffered is ‘X’. This method is standard in proving lower bounds in bandits (for example, see Chapter 2 of Introduction to Multi-armed Bandits by Alexander Slivkins).
>
> Reviewer's concern: “Could they also explain how does the single-pass restriction appear in the proof?”
>
> Author's response:  Observations 1 and 2 hold only under the single-pass setting since the single-pass setting does not allow an arm to be read back into the arm-memory once it has been discarded. We have explained the single-pass setting in detail in Lines 145-152. Additionally, we clearly state in lines 102-105 that the $O(\sqrt{T})$ regret upper bounds in the previous works are for the multi-pass setting whereas we look at the more challenging single-pass setting, which was mentioned to be an important open problem in the previous work, and this leads to this gap.
>
>
>
> Reviewer's concern: “Could the authors explain why observations 1 and 2 hold?”
>
> Author's response: Observation 1 and 2 directly follows from the definitions, as follows:
>
> Observation 1: The event  $\bar{A_1}$ is the set of all reward realizations such that the first time step at which some arm from 1 to m is discarded from the memory $T’_{\omega}$ is greater than L. Under instance I_0, the first m arms have mean ½ whereas the best arm has mean 1. Hence, the regret in the first L rounds is exactly (1-1/2)L. Substituting the value of L gives us the observation.
>
> Observation 2: The event $A^{i'}$ is the set of all outcomes such that $T’_{\omega}$, the first time step at which some arm from arm_1 to arm_m is discarded from the memory, is less than L and arm $i'$ is one of the discarded arms. In instance $I_{i'}$, arm $i'$ is the optimal arm and has a reward gap $\varepsilon/2$ from the other arms. Hence, discarding it within L rounds leads to some sub-optimal arm being pulled in the remaining T-L rounds. Given the choice of L, this leads to the regret mentioned in Observation 2.
>
> We omitted these trivial proofs due to space limitations.
>
>
> Reviewer's concern: “In fact, one can note that finding a minimax optimal algorithm that is δ-PAC is an ill-posed problem since one can easily show that the lower bound in the minimax sense is infinite. Perhaps the authors should look at the paper entitled "Optimal Best Arm Identification with Fixed Confidence" by Garivier and Kaufmann.”
>
> Author's response: We study the $\varepsilon$-best arm identification problem and not the optimal best-arm identification problem. The $\varepsilon$-best arm identification problem with $\delta$-PAC guarantee has been shown to have a sample complexity lower bound of $\Omega(n/\varepsilon^2 \log(1/\delta))$ [1]. [1] also proposes the Median-Elimination algorithm with matching (optimal) sample complexity. Also, note that the lower bound in [1] which does not have any restriction on the arm-memory naturally translates to a lower bound in our setting. Our algorithm provides optimal sample complexity in this sense while having arm-memory size of $O(\log^* n)$, which is nearly a constant.
>
> [1] Even-Dar, Eyal, Shie Mannor, and Yishay Mansour. "PAC bounds for multi-armed bandit and Markov decision processes." International Conference on Computational Learning Theory. Springer, Berlin, Heidelberg, 2002.
>
>
> Reviewer's concern: “The arrival of arms is poorly explained. Can the authors clarify how do arms arrive in detail? is there a delay between the arms' arrival? is there a probability distribution that governs the arms' arrival? Shouldn't that impact, for instance, the regret lower bound, if the best arm has a low probability of arrival?”
>
> Author's response: Our model has been previously studied by RoyChaudhuri and Kalyanakrishnan [AAAI’20] (citation 24 in our submission), Assadi and Wang [STOC’20] (citation 3 in our submission), etc. The functioning of the algorithm is described in detail on Lines 51-62.
>
> One can think of the arm arrival as follows:  Let the arms be indexed from 1 to n. Let j be a counter that keeps track of the index of the next arm in the sequence to be considered to be read into the memory. Before the algorithm begins execution, j is initialized to 1. At the beginning of any time step $t\leq T$, the algorithm may decide to read arm j into the arm-memory and then increment j by 1 or may opt to not read any arm into arm-memory at the time step t. The algorithm can read multiple arms at the start of any time step (by continuously incrementing j), as long as the number of arms in the arm-memory is at most m. At each time step t, the algorithm samples one of the arms currently in the arm-memory.
>
>
>
> Reviewer's concern: “Existing regret upper bounds have a worst-case regret upper bound of order  $O(\sqrt{T})$, while the provided lower bound is of order $\Omega(T^{3/2})$ . It can be understood from the authors' result that this is because the class of algorithm over which the lower bound holds is restricted to a single-pass algorithm. Could the author confirm this and provide a discussion on this?”
>
> Author's response:  Yes, our lower bound of order $\Omega(T^{3/2})$ is for single-pass algorithm when we are allowed to store at most n-1 arms in the arm-memory, whereas one can achieve $O(\sqrt{T})$ upper bound  by UCB (which can viewed as a single-pass algorithm where we are allowed to store all n arms in the arm-memory). This is mentioned in Lines 70-78 in our submission.  Also $O(\sqrt{T})$ upper bound is achieved with limited arm-memory by a $\log T$ pass algorithm by RoyChaudhuri and Kalyanakrishnan [AAAI2020] (citation 24 in our submission). Obtaining  $O(\sqrt{T})$ upper bound with limited arm-memory using less than log T passes is open and could be an interesting future work. This is mentioned in Lines 111-113 in our submission.

---

### Decision · Program_Chairs · 2021-09-27

**Decision:**

Accept (Poster)

**Comment:**

The committee was divided on this paper and the discussions have highlighted that the despite several writing issues, the paper presents novel and important contributions. We suggest the authors to revise their draft for clarity but we recommend to accept this work to Neurips 2021.